# Motoneuron-driven computational muscle modelling with motor unit resolution and subject-specific musculoskeletal anatomy

**Arnault H. Caillet[1,2], Andrew T. M. Phillips[1], Dario Farina[2‡]*, Luca Modenese[1,3‡]***

**1** Department of Civil and Environmental Engineering, Imperial College London, London, United Kingdom, **2** Department of Bioengineering, Imperial College London, London, United Kingdom, **3** Graduate School of Biomedical Engineering, University of New South Wales, Sydney, Australia

‡ These authors are joint senior authors on this work.
* d.farina@imperial.ac.uk (DF); l.modenese@unsw.edu.au (LM)

**Data Availability Statement:** The code used to generate the results is available at the following public GitHub repository: https://github.com/ArnaultCAILLET/MN-driven-Neuromuscular-

## Abstract

The computational simulation of human voluntary muscle contraction is possible with EMG-driven Hill-type models of whole muscles. Despite impactful applications in numerous fields, the neuromechanical information and the physiological accuracy such models provide remain limited because of multiscale simplifications that limit comprehensive description of muscle internal dynamics during contraction. We addressed this limitation by developing a novel motoneuron-driven neuromuscular model, that describes the force-generating dynamics of a population of individual motor units, each of which was described with a Hill-type actuator and controlled by a dedicated experimentally derived motoneuronal control. In forward simulation of human voluntary muscle contraction, the model transforms a vector of motoneuron spike trains decoded from high-density EMG signals into a vector of motor unit forces that sum into the predicted whole muscle force. The motoneuronal control provides comprehensive and separate descriptions of the dynamics of motor unit recruitment and discharge and decodes the subject's intention. The neuromuscular model is subject-specific, muscle-specific, includes an advanced and physiological description of motor unit activation dynamics, and is validated against an experimental muscle force. Accurate force predictions were obtained when the vector of experimental neural controls was representative of the discharge activity of the complete motor unit pool. This was achieved with large and dense grids of EMG electrodes during medium-force contractions or with computational methods that physiologically estimate the discharge activity of the motor units that were not identified experimentally. This neuromuscular model advances the state-of-the-art of neuromuscular modelling, bringing together the fields of motor control and musculoskeletal modelling, and finding applications in neuromuscular control and human-machine interfacing research.

Model-with-motor-unit-resolution. The segmented medical images and the subject-specific MSK model are available at the following Zenodo repository: https://zenodo.org/records/10069266. The code used to reconstruct the complete motor unit pool from experimental motoneuronal data is available at: https://github.com/ArnaultCAILLET/Caillet-et-al-2022-PLOS_Comput_Biol.

**Funding:** This work was supported by Imperial College London (Skempton Scholarship to AHC), the European Research Council (810346 to AHC) and the University of New South Wales (Scientia Fellowship to LM). The funders had no role in study design, data collection and analysis, decision to publish, or preparation of the manuscript.

**Competing interests:** The authors have declared that no competing interests exist.

## Author summary

Neuromuscular computational simulations of human muscle contractions are typically obtained with a mathematical model that transforms an electromyographic signal recorded from the muscle into force. This single-input single-output approach, however, limits the comprehensive description of muscle internal dynamics during contraction because of necessary multiscale simplifications. Here, we advance the state-of-the-art in neuromuscular modelling by proposing a novel mathematical model that describes the force-generating dynamics of the individual motor units that constitute the muscle. For the first time, the control to the population of modelled motor units was inferred from decomposed high-density electromyographic signals. The model was experimentally validated, and the sensitivity of its predictions to different experimental neural controls was assessed. The neuromuscular model, coupled with an image-based musculoskeletal model, includes a novel and advanced neuromechanical model of the motor unit excitation-contraction properties, and is suited for subject-specific simulations of human voluntary contraction, with applications in neurorehabilitation and the control of neuroprosthetics.

## Introduction

During voluntary skeletal muscle contractions, the pool of alpha-motoneurons (MNs) and the muscle fibres they innervate, constituting the muscle's population of motor units (MUs), transform in a series of chemical-mechanical events the neural message from the spinal cord into molecular forces [1]. These forces sum across the multiscale architecture of the muscle to build the individual MU forces and the whole muscle force. Extensive experimental investigations in animals *in vivo* have advanced our understanding of the neuromechanical dynamics responsible for voluntary muscle contraction. For example, the force generated by skeletal muscles is modulated by the discharge frequencies of the innervating MNs [2], and by the number of discharging MUs, which are sequentially recruited in the increasing order of their size and the force they can generate [3,4]. Because invasive *in vivo* practices in humans are reduced to specific electrode insertions [5,6] or challenging surgeries [7–9], our understanding of human voluntary muscle contraction mainly relies on non-invasive *in vivo* techniques, such as bipolar (bEMG) and high-density (HDEMG) surface electromyography [10–12], imaging [13,14], and joint torque recordings. These non-invasive techniques are however challenging to apply during real-time contractions on multiple muscles, provide limited information that is usually not easily interpreted, or are commonly constrained to simple tasks like isometric contractions. When some neuromuscular properties and dynamics cannot be measured in humans, they are often extrapolated from animals, despite important limitations [9,15,16].

To support experimental investigations into understanding and simulating the interplay between human motor control and force generation, mathematical models of skeletal muscles were developed to mimic the dynamics of muscle contraction. Hill-type neuromuscular models are popular solutions which have been extensively reviewed recently [17]. They rely on few parameters, are computationally cheap and conceptually simple, while complex enough to describe the chain of neuromechanical events responsible for muscle force generation, which makes them accurate, flexible and capable of refinement [18]. Hill-type models have been used to explore voluntary neuromuscular control with EMG-driven forward simulations of muscle contraction [19–23]. With these methods, the recorded EMG signals are processed to provide a unique neural control to a single Hill-type actuator that phenomenologically describes whole

muscle dynamics. This approach is well-established and implemented in automated tools [21] and has had important applications, for example in human-machine interfacing [24–27].

However, and despite attempts to address this limitation [28–32], this single-input approach lumps the recruitment and discharge dynamics of the muscle's individual MUs into a unique phenomenological neural control for the whole MU population, the resulting force-generating events of which are hence lumped into single representative quantities for the whole MU pool. These representative quantities usually overlook the continuous distribution of the MUs' neuromechanical properties across the MU pool and are difficult to calibrate and interpret [32]. Consequently, this single-actuator macroscopic approach does not provide an adequate structure to physiologically describe the independent and inter-related [33–35] dynamics of a muscle's MU pool. These limitations hinder the investigation of human neuromuscular control with computational tools. In response to this limitation, other studies developed more physiological muscle models described as populations of Hill-type actuators, each of which simulated the dynamics of individual MUs [36–38]. Yet, those models received synthetic motoneuronal signals and were not tested, and most of them not validated in conditions of voluntary neuromuscular control. Recent advances in the recording and decomposition of surface HDEMG signals [39,40] allow the non-invasive identification of trains of action potentials for large samples of MUs discharging during human voluntary contraction. To date, those vectors of identified MU spike trains were systematically compiled into single neural controls to drive single Hill-type muscle actuators [22,23]. Consequently, the experimental spiking behaviour of the identified discharging MNs was never used to control Hill-type-like models of individual MUs in the simulation of human voluntary contraction.

To address this gap, in this study a novel MN-driven neuromuscular model was developed and validated. The model was described as a population of Hill-type actuators, each of which simulated the neuromechanical and force-generating dynamics of individual MUs. In forward simulations of human voluntary contraction, the MN-driven model transformed an experimental vector of MN spike trains decoded from HDEMG signals [40], that comprehensively described the dynamics of MU recruitment and rate coding, into a vector of simulated MU forces that summed into the predicted whole muscle force. Sampling the muscle into individual MUs provided an adequate structure for proposing robust multiscale simplifications, advanced models of the MU activation dynamics, and physiological muscle-specific distributions of the MU's neuromechanical properties across the MU pool, that were scaled to subject-specific values derived from a subject-specific musculoskeletal (MSK) model. It was shown that accurate muscle force predictions were obtained when the experimental motoneuronal controls accurately described the real discharge activity of the MU pool. This was obtained with experimental samples of MN spike trains derived from large and dense grids of EMG electrodes, and, to some extent, when the discharge activity of the complete MU pool was reconstructed from experimental data with computational methods [41]. By proposing novel solutions for controlling and designing Hill-type-like neuromuscular models, this study advances the state-of-the-art of neuromuscular modelling, opens avenues for investigating the interplay between the central nervous system and the neuromuscular machinery during human voluntary contraction, reconciles the complementary fields of motor control and MSK modelling, and finds applications in numerous fields, including the investigation of the human neuromuscular dynamics and neural synergies [35] during voluntary contractions, and the design and control of neuroprosthetics [42,43]. The implementation of the method is publicly available at https://github.com/ArnaultCAILLET/MN-driven-Neuromuscular-Model-with-motor-unit-resolution. The segmented medical images and the subject-specific MSK model are publicly available at https://zenodo.org/records/10069266.

## Methods

### 1. Overview of MN-driven neuromuscular modelling

We developed an experimental and computational method (Fig 1) for simulating, with a MN-driven neuromuscular model, the neuromechanical dynamics of a subject-specific population of individual MUs during human voluntary isometric muscle contraction. From surface HDEMG signals recorded on the Tibialis Anterior (TA) muscle (Fig 1A and 1B), we decoded the experimental discharge activity of the population of identified MUs (Fig 1C), which was extended to the complete MU pool (Fig 1D) using an open-source computational method [41], available at https://github.com/ArnaultCAILLET/Caillet-et-al-2022-PLOS_Comput_Biol. The accuracy of both the experimental and reconstructed populations of spike trains $sp_i(t)$ in estimating the neural drive to muscle $\overline{D}(t)$ was validated (Fig 1E). We described the MN-driven neuromuscular model (Fig 1F) as a collection of in-parallel Hill-type Force Generators (FGs), each of which was controlled by a dedicated spike train $sp_i(t)$. Each FG described the neuromechanics of a MU and included a Neuromechanical Element (NE) and a Contractile Element (CE) to describe the excitation-contraction coupling properties (MU excitation and activation dynamics) and mechanical properties (MU contraction dynamics) of the MU, respectively [17]. The FGs individually transformed the spike trains $sp_i(t)$ into the MU forces $f_i^{MU}(t)$ that collectively generated the whole muscle force $F^M(t)$ (Fig 1G). We further scaled the MN-driven neuromuscular model to be subject-specific with muscle-tendon properties derived from a subject-specific MSK model obtained from segmented magnetic resonance images (MRIs) (Fig 1H and 1I). We finally validated the predicted force $F^M(t)$ against a reference muscle force $F_{TA}(t)$ (Fig 1G), that was estimated (Fig 1J) from the recorded ankle torque $T(t)$ and the bipolar EMG (bEMG) activity of the TA's agonist and antagonist muscles crossing the ankle joint.

In the following, when subject-specific properties could not be measured *in vivo*, the coefficients of the mathematical equations describing the model were tuned with experimental data from the literature, considering in turn and decreasing order of preference experimental studies on individual human MUs or bundles of fibres, human fibres and sarcomeres, cat fibres, rodent fibres, rodent muscles, and amphibian fibres or muscles. Any experimental quantity measured at the sarcomere and/or fibre scales was directly assigned to the MU scale.

### 2. Experimental data and subject-specific neuromusculoskeletal quantities

**2.1. Contraction tasks, force and EMG recordings, and subject-specific spiking activity.** The experimental approach represented in Fig 1A was described elsewhere [40] and is briefly summarized here. One healthy male participant (age: 26 yr; height: 168 cm; body weight: 52 kg) volunteered to participate in the experimental session of the study. The participant sat on a massage table with the hips flexed at 30˚, 0˚ being the hip neutral position, and his knees fully extended and strapped to the massage table to avoid any knee and hip motion and co-contraction of the thigh muscles during the ankle dorsiflexion. The ankle torque signals $T(t)$ were recorded with a load cell (CCT Transducer s.a.s, Turin, Italy) connected in-series to the pedal of a commercial dynamometer (OT Bioelettronica, Turin, Italy), to which the participant's foot was fixed at 30˚ in the plantarflexion direction, 0˚ being the foot perpendicular to the shank. The effects of gravity, including the rig weight, were cancelled by removing the initial measured offset during the experiments. After registering the participant's maximum voluntary contraction (MVC) in dorsiflexion, the participant performed two trapezoidal isometric contractions at 30% and 50% MVC with 120 s of rest in between, consisting of linear ramps up and down performed at 5%/s and a plateau maintained for 20 s and 15 s at 30% and 50% MVC, respectively.

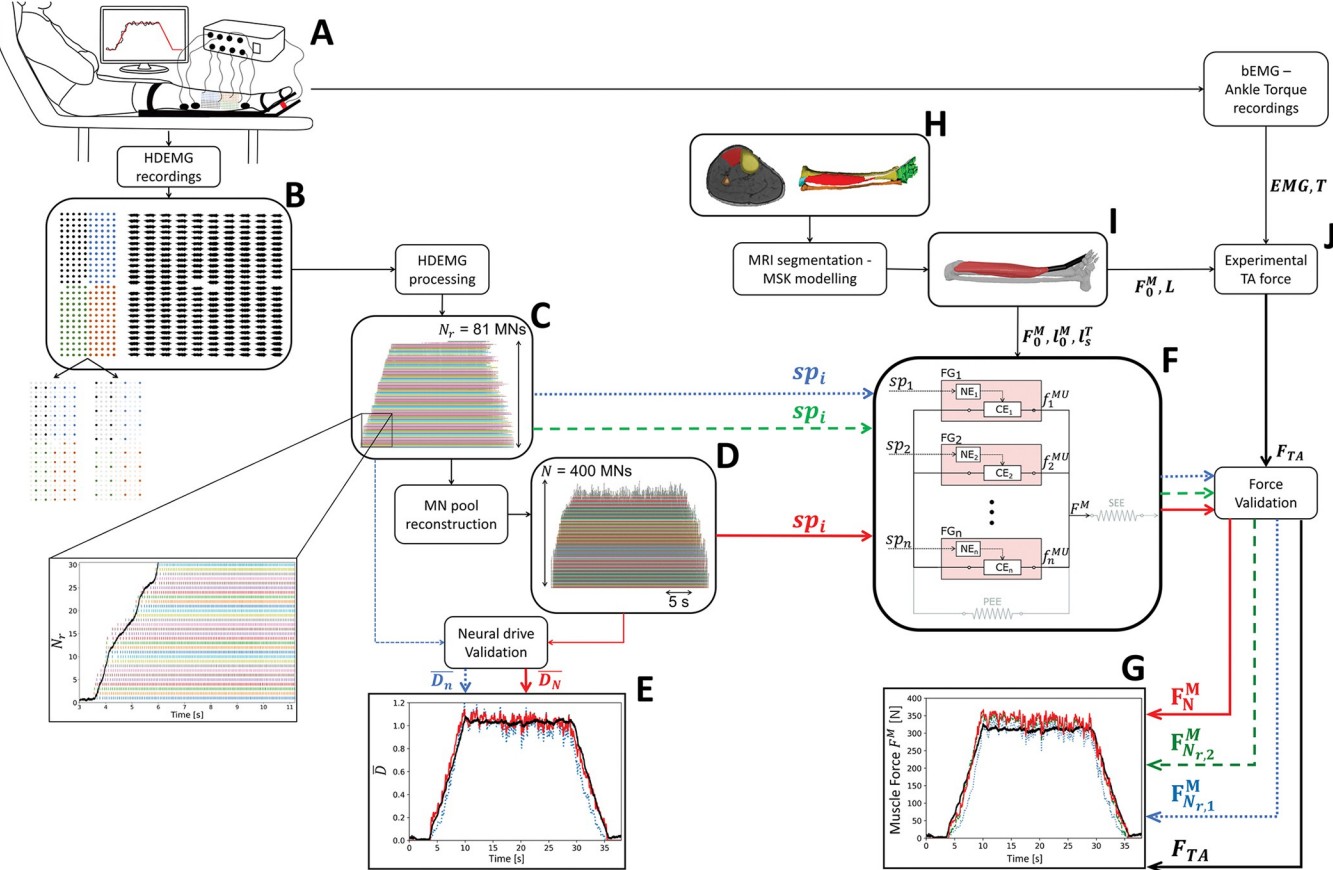

**Fig 1. Description of the steps involved in the forward prediction and validation of the TA muscle force during isometric contractions using the developed subject-specific MN-driven neuromuscular model.** During trapezoidal isometric ankle dorsiflexions (A), HDEMG signals were recorded from the TA muscle with a grid of 256 EMG electrodes, built with four grids of 64 electrodes (B), which was artificially downsampled into two grids of lower electrode density. Using convolutive blind-source separation, the HDEMG signals were decomposed into $N_r$ identified motor unit spike trains (C), from which the discharge activity of the complete MU pool was inferred and reconstructed (D) [41]. The experimental and reconstructed populations of spike trains $sp_i(t)$ were validated for their accuracy in estimating the normalized effective neural drive $\overline{D}(t)$ to muscle (E). The MN-driven neuromuscular model developed in this study (F), consists of a collection of in-parallel Hill-type MU Force Generators (FGs), each including a Neuromechanical Element (NE) and a Contractile Element (CE), that collectively generate the whole TA muscle force $F^M$. $F^M$ was predicted (G) from the experimental $sp_i(t)$ trains in a blind approach (blue dotted lines), after location of the identified MUs into the MU pool (green dashed lines), and from the completely reconstructed population of spike trains (solid red lines). The neuromuscular model was scaled with subject-specific muscle-tendon properties derived from a subject-specific MSK model (I), obtained from segmented MRI scans (H). The predicted TA force $F_{N_{r,1}}^M$, $F_{N_{r,2}}^M$, or $F_N^M$, depending on the type of neural control, was finally validated (G) against the experimental TA force $F_{TA}$ (J). $F_{TA}$ was estimated from the recorded ankle torque $T$ and the estimated co-activity of the TA's agonist and antagonist muscles crossing the ankle joint.

To derive subject-specific motoneuronal controls to the neuromuscular model, HDEMG signals were recorded over the TA muscle during a first experimental session with a total of 256 EMG electrodes (Fig 1B), distributed between four rectangular grids of 64 electrodes, with 4 mm interelectrode distance covering 36 cm$^2$ of the muscle surface (10 cm x 3.6 cm; OT Bioelettronica). HDEMG and force signals were recorded using the same acquisition system (EMG-Quattrocento; OT Bioelettronica) with a sampling frequency of 2,048 Hz. As previously described [40], the 256-electrode grid was downsampled by successively discarding rows and columns of electrodes and artificially generating two new grids of lower electrode density (Fig 1B). The two grids covered the same muscle area with interelectrode distance of 8 mm and 12 mm, involving 64 and 36 electrodes, respectively. After visual inspection and band-pass filtering, the HDEMG signals recorded with the three grids were decomposed into MUs spike trains

using convolutive blind-source separation, as previously described [39]. After the automatic identification of the MUs duplicates were automatically removed according to a threshold of 30% of discharge times in common between identified spike trains [40], and all the MUs spike trains were visually checked for false positives and false negatives [11]. Only the MUs which exhibited a pulse-to-noise ratio > 28 dB were retained for further analysis. The final $N_r$ spike trains $sp_i(t)$ identified experimentally (Fig 1C) were stored as binary vectors of zeros and ones identifying the sample times when a discharge occurred. The ankle torques $T^{th}$ at which the $N_r$ MUs were recruited, also called *torque recruitment thresholds* in the following, were calculated as the average of the recorded $T$ values over a 10 ms range centred around the MUs' first identified discharge time. In the following, the $N_r$ MUs identified experimentally were ranked in the ascending order of measured recruitment torques $T_i^{th}$ with the index $i \in [\![1; N_r]\!]$. According to the Henneman's size principle [3,4], these MUs were therefore also ranked in the same order of current recruitment threshold $I_{th}$, maximum isometric forces $f_0^{MU}$, and innervation ratios $IR$ according to Eq 1.

$$\forall j, k \in [\![1; N_r]\!], j < k \Longleftrightarrow I_{th}(j) < I_{th}(k) \Longleftrightarrow T^{th}(j) < T^{th}(k)$$
$$\Longleftrightarrow f_0^{MU}(j) < f_0^{MU}(k) \Longleftrightarrow IR(j) < IR(k) \tag{1}$$

In a second session where the isometric contractions were repeated following the same protocol, bEMG signals were recorded from the pair of agonist extensor digitorum longus (EDL) and extensor hallucis longus (EHL) muscles, and from the antagonist Gastrocnemius Medialis (GM) and Lateralis (GL), and Soleus (SOL) muscles with adhesive bipolar electrodes (OT Bioelettronica). During this second session, the participant additionally produced two MVCs in plantarflexion. The bEMG signals were band-pass filtered (10–450 Hz), full-wave rectified, and then low-pass filtered (2 Hz) [22], before the resulting envelopes were normalized with respect to the peak processed EMG values obtained in the MVC trials.

**2.2. MRI scans, subject-specific MSK quantities, and experimental force estimation.** To develop a subject-specific model of the MSK system to couple with the neuromuscular model, a 3D T1-weighted VIBE (volumetric interpolated breath-hold examination) sequence was used to acquire high resolution images (0.45 x 0.45 mm pixel size, 1 mm slice thickness and increment) from the knee joint to mid-foot of the participant's dominant leg. The sequence consisted in three blocks with 50 mm overlap, during which the participant was asked not to move to facilitate a successful merging of the adjacent blocks.

The volumetric shapes of the tibia, fibula, and foot bones, and of the TA muscle belly and tendon were carefully identified by manual segmentation (Fig 1H) using ITK-SNAP [44] and an anatomical atlas as support [45]. Slices of the agonist EDL and EHL and antagonist SOL, GM, and GL muscle bellies were segmented at regular intervals along the tibial length so that their centroidal lines of action could be outlined. All the main tendons crossing the ankle joint could be clearly identified and were fully segmented. An expert radiologist confirmed the accuracy of the segmentation. The segmented shapes were manually adjusted on the transversal plane to avoid inconsistency of the muscle geometry in correspondence of the MRI sequence blocks using Netfabb (https://www.autodesk.com/products/netfabb/) and Meshlab [46]. The TA muscle volume $V^M$ computed from the segmentation was found consistent with the relationships between anthropometry and muscle volumes proposed in [47] from a cohort of segmented MRI scans. Therefore, the volume $V^M$ of the EDL, EHL, SOL, GM, GL muscle bellies was estimated with the volume-height-mass relationship provided in the supplementary material of [47].

A subject-specific skeletal model of the ankle joint compatible with OpenSim (version 4.4) [48,49] was automatically generated from the segmented bone volumes (Fig 1I) using the

open-source automated toolbox STAPLE [50]. With this approach, the ankle joint was modelled as a hinge joint, the axis of rotation of which was personalised based on the shape of the articular surfaces of the segmented talus bone. Using another automated approach [51], the centroidal line of action of TA was automatically computed from the segmented muscle geometry. Similarly, the centroidal lines of action for EDL, EHL, SOL, GM, and GL were manually outlined from the segmented muscle slices. The muscle tendons paths were manually defined in NMSBuilder [52] with an approach previously proposed [53] that uses the segmented tendons as reference.

To reproduce the experimental conditions, the plantarflexion angle of the final subject-specific MSK model was set to 30˚. At this angle, the subject-specific muscle-tendon length $l^{MT}$ of the six muscles and their moment arm $L$ with respect to the ankle joint were computed using OpenSim. The subject-muscle-specific optimal length $l_0^M$ and the tendon slack length $l_s^T$ of the six muscles were estimated from the muscle-specific values $l_{0,Raj}^M$ and $l_{s,Raj}^T$ proposed in a generic published model [54], which were scaled with Eq 2 by maintaining their ratio with respect to the entire musculotendon lengths $l_{Raj}^{MT}$ and $l^{MT}$ in the neutral ankle position.

$$\begin{cases} l_0^M = \dfrac{l^{MT}}{l_{Raj}^{MT}} \cdot l_{0,Raj}^M \\[3mm] l_s^T = \dfrac{l^{MT}}{l_{Raj}^{MT}} \cdot l_{s,Raj}^T \end{cases} \tag{2}$$

Considering a specific tetanic tension of $\sigma = 60$ $N/cm^2$ [54], the subject-specific maximum isometric force $F_0^M$ of the six muscles was obtained from the known muscle volumes $V^M$ with Eq 3.

$$F_0^M = \frac{V^M}{l_0^M} \cdot \sigma \tag{3}$$

Combining these subject-muscle-specific properties with the bEMG signals recorded from the co-contracting flexor muscles during the second experimental session (Fig 1A), the experimental force $F_{TA}(t)$ developed by the TA during the first experimental session was inferred (Fig 1J) from the measured experimental ankle torque $T(t)$ with Eq 4. In Eq 4, $L_{TA}$ is the moment arm the TA tendon makes with the ankle joint, and $\Delta T(T)$ relates the level of measured ankle torque $T(t)$ to the algebraic amount of ankle torque $\Delta T$ taken by the group of agonist EHL and EDL and antagonist SOL, GM, and GL muscles. The continuous $\Delta T(T)$ relationship was obtained by trendline fitting the $(T; \Delta T)$ cloud of points measured during the second experimental session. Details on the calculation of $\Delta T(T)$ are provided in S1 Text (Section 1).

$$F_{TA}(t)[N] = \frac{T(t) - \Delta T(T(t))}{L_{TA}} \tag{4}$$

## 3. Generic properties of the MU pool in the human TA muscle

From an extensive review of the experimental human TA literature performed elsewhere [41], it was assumed that a typical adult TA muscle included $N_f^{tot} = 200,000$ muscle fibres gathered into $N = 400$ MUs. This review also returned that the TA MU torque recruitment thresholds $T^{th}$, expressed in percentage of the maximum recorded torque (% MVC), followed the generic linear-exponential continuous distribution in Eq 5 and displayed in Fig 2A. In Eq 5, $j$ is an integer identifying the $j^{th}$ MU in the entire pool ranked in the ascending order of MU torque

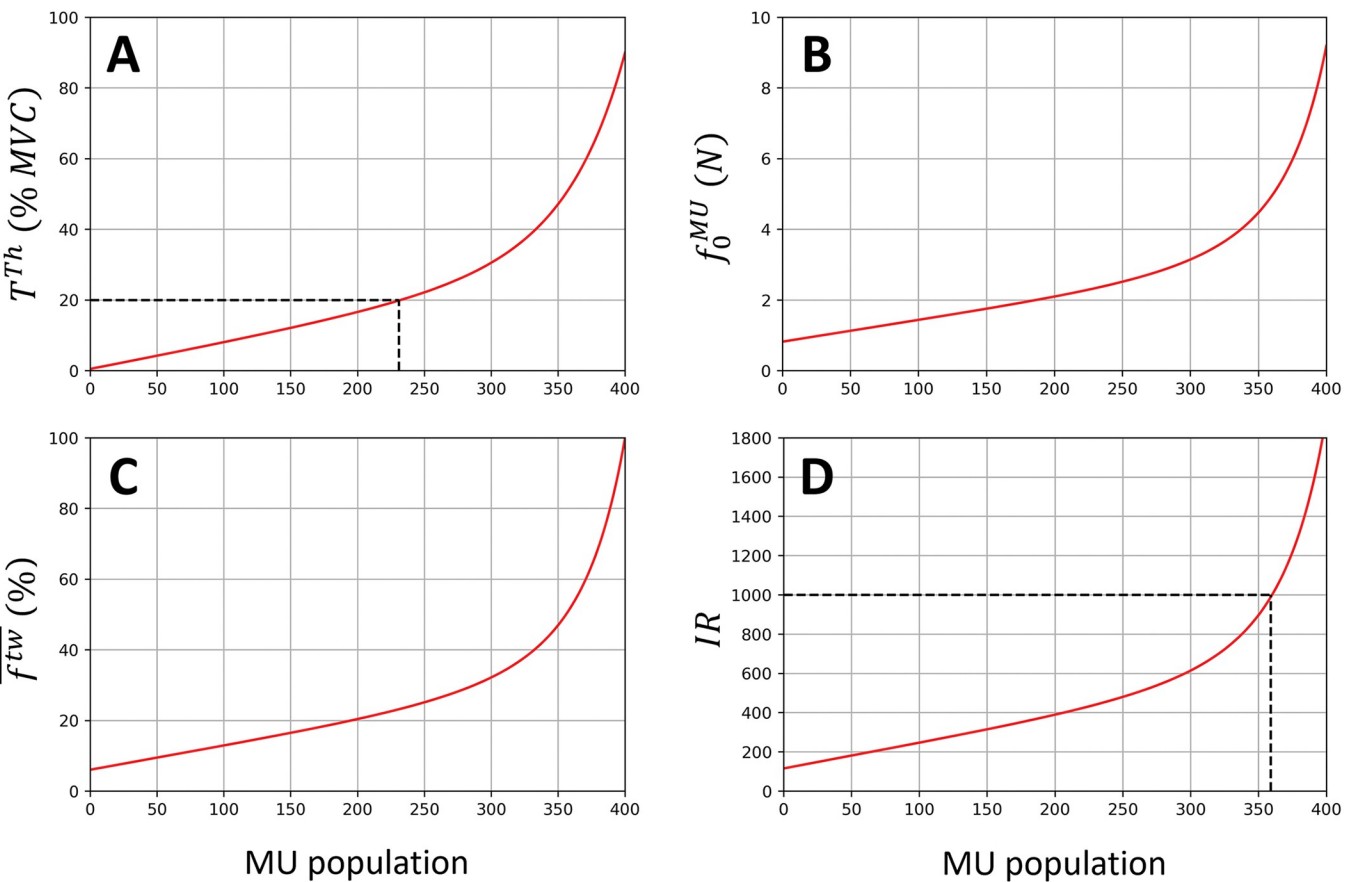

**Fig 2.** In a generic human TA muscle, distributions in the pool of $N = 400$ MUs of the MU torque recruitment thresholds $T^{th}$ (A), MU maximum isometric forces $f_0^{MU}$ (B), MU twitch forces $\overline{f^{tw}}$ normalized to the muscle maximum isometric force $F_0^M$ and expressed in percentage (C), and MU innervation ratios $IR$ (D), calculated with Eq 5 to Eq 9. (A) As identified by the dashed line, 231 TA MUs, i.e., 58% of the MU pool is recruited below 20% MVC, which is consistent with previous conclusions [2]. (B) The normalized distribution in Eq 8 was scaled to Newtons with the subject-specific maximum isometric force of the TA $F_0^M$ derived previously. From the subject-specific moment arm of the TA derived previously, TA MUs are expected to produce 20 to 230·1$^{-3}$ Nm twitch torques with this distribution, which is consistent with the measurements reported in [5]. (D) As identified by the dashed line, the 359 smallest TA MUs gather 72% of the fibre constituting the muscle and are assumed slow-type MUs. According to Eq 1, the MUs are ranked in the same ascending order of $F^{th}, f_0^{MU}, \overline{f^{tw}}$, and $IR$.

recruitment thresholds.

$$T^{th}(j) = 0.50 \cdot \left(58.12 \cdot \frac{j}{N} + 120^{\left(\frac{j}{N}\right)^{1.83}}\right), j \in [\![1; N]\!] \tag{5}$$

The $N_r$ MUs identified in the experimental trials were ranked with the index $i$ ($i \in [\![1; N_r]\!]$, with $N_r < N$) in the ascending order of measured $T_i^{th}$. Using the $T^{th}(j)$ distribution in Eq 5, these experimental MUs were mapped into the complete $T^{th}$-ranked pool of 400 MUs, where they were assigned a new index $N_i$ ($N_i \in [\![1; N]\!]$), as shown in Eq 6. To do so, the equation $T_i^{th} = T^{th}(N_i)$ was solved for the unknown $N_i$ for the $N_r$ experimental MUs.

$$i \in [\![1; N_r]\!] \rightarrow N_i \in [\![1; N]\!] \tag{6}$$

Never measured in human muscles *in vivo*, the continuous distribution of the normalized MU maximum isometric forces $\overline{f_0^{MU}}(j)$ in the TA was here approximated from a continuous distribution $\overline{f^{tw}}(j)$ of MU twitch force normalized to the muscle maximum isometric force $F_0^M$.

Processing published experimental TA data [5,55,56] with the same method as used for Eq 5, Eq 7 was obtained. In Eq 7, a 16.3-fold range was taken for the $f^{tw}$ values, consistently with Fig 3A in [5] for a single subject, as opposed to a 250-fold range proposed in a recent review [57] that relied on the same figure, but considering unnormalized merged data from ten subjects.

$$\overline{f^{tw}}(j) = 6.07 \cdot \left(4.52 \cdot \frac{j}{N} + 11.96^{\left(\frac{j}{N}\right)^{4.66}}\right), \; j \in [\![1;N]\!] \tag{7}$$

The normalized $\overline{f_0^{MU}}(j)$ distribution in Eq 8 was then estimated by scaling the coefficients in Eq 7 to obtain a 11-fold range for the $\overline{f_0^{MU}}$ values in the TA MU pool, to account for the different twitch-tetanus ratios obtained between slow and fast fibres in the rodent literature [58]. In Eq 8, $\sum_{j=1}^{N} \overline{f_0^{MU}}(j) = 1$. The $f_0^{MU}(j)$ distribution in Newtons, displayed in Fig 2B, is finally obtained by multiplying Eq 8 by the $F_0^M$ value obtained with Eq 3.

$$\overline{f_0^{MU}}(j) = 7.86 \cdot 10^{-4} \cdot \left(3.00 \cdot \frac{j}{N} + 8.20^{\left(\frac{j}{N}\right)^{5.29}}\right), \; j \in [\![1;N]\!] \tag{8}$$

Human muscles also display a continuous distribution of slow-to-fast MU types in the MU pool. It is known that, in a human TA of $N = 400$ MUs, 72% of the $N_f^{tot} = 200{,}000$ TA fibres are Type 1 'slow' fibres [59]. Considering that the innervation ratio $IR$ of a MU is roughly proportional to the amplitude of the MU force twitch $f^{tw}$ [4], Eq 9 gives the continuous distribution of $IR$, displayed in Fig 2D.

$$IR(j) = \frac{\overline{f^{tw}}(j)}{\sum_{k=1}^{400} \overline{f^{tw}}(k)} \cdot N_f^{tot}, \; j \in [\![1;N]\!] \tag{9}$$

Although it is debated [4] whether the MU type, currently determined by the MU twitch force $f^{tw}$, is also correlated or not to the MU size, i.e., to the MU Force recruitment threshold in humans, it was here assumed that the small low-threshold MUs were of the 'slow' type, and the large high-threshold MUs of the 'fast' type. Therefore, from the cumulative summation of the MU $IRs$, the 359 lowest-threshold MUs that have the lowest innervation ratios were considered to account for 72% of $N_f^{tot}$ and to be of the 'slow' type in the following.

## 4. Development of a MN-driven neuromuscular model described as a population of Hill-type MU actuators

**4.1. Rheological description and preliminary simplifications.** The MN-driven neuromuscular model developed in this study (Fig 1F) consists of a population of $n$ in-parallel FGs, that describe the force-generating activity of the MU pool with mathematical models of the individual MU's force-generating dynamics (Fig 3). $n$ is the length of the vector of available input spike trains controlling the model and takes the values $N_r$ or $N$ in this study. Using step-by-step simplifications justified in detail in S1 Text (Section 2), {1} the TA tendon was assumed rigid (tendon length set to the tendon slack length $l_s^T$ [60]) and the in-series elastic element (SEE, in grey in Fig 1F) and its passive force dynamics were neglected, {2} all the MU actuators were assigned the same optimal length, set as the muscle's $l_0^M$, and the same normalized constant length, set as $\bar{l} = 1.16$ according the subject-specific MSK measurements, {3} the Force-Velocity properties of the MUs were therefore neglected, and {4} the passive force of the CE was found negligible, so that the passive elastic element (PEE, in grey in Fig 1F) was neglected. With these assumptions, and after normalizing the force $F^M$ and MU length $l$ state variables to the $F_0^M$ and $l_0^M$ parameters and reporting them with a bar, as described in S1 Text (Section 2),

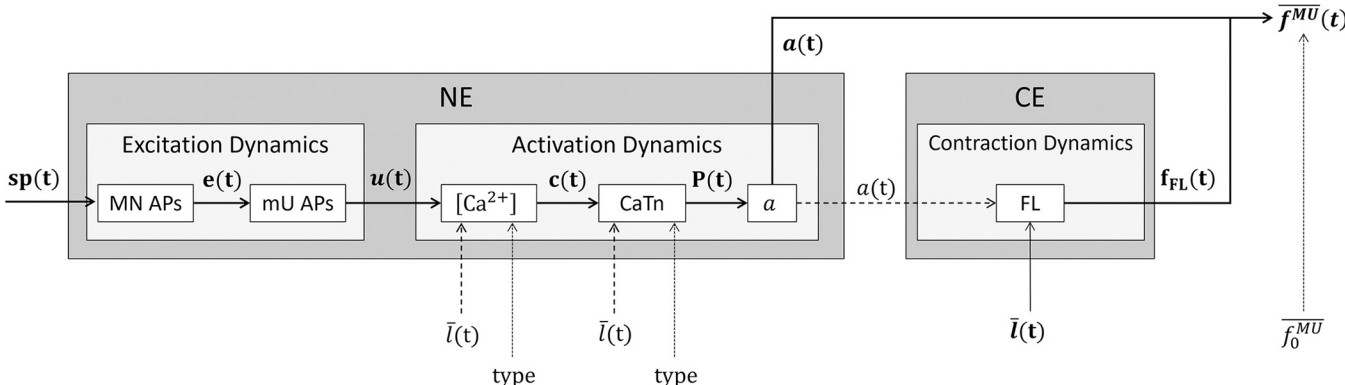

**Fig 3. For a representative motor unit (MU) of the pool of Hill-type MUs in Fig 1F, description of the cascading dynamics that transform an input spike train $sp(t)$ into a normalized MU force $\overline{f^{MU}}(t)$.** The time-history of $\overline{f^{MU}}(t)$ is obtained with Eq 10 by product of the scaling factor $f_{FL}(t)$, the MU active state $a$ $(t)$, and the MU normalized isometric force $\overline{f_0^{MU}}$. $f_{FL}$ is obtained with a normalized FL relationship (Fig 4A) that entirely describes the Contraction Dynamics of the Contractile Element (CE) in isometric conditions. The Neuromechanical Element (NE) is controlled by the binary spike train $sp(t)$, which predicts during the MU's Excitation Dynamics the trains of motoneuron (MN) and muscle Unit (mU) Action Potentials (APs) in Volts (Fig 4B), identified with the $e(t)$ and $u$ $(t)$ state variables, respectively. The discharge activity $u(t)$ triggers the MU's activation dynamics by successively controlling, with the $c(t)$, $P(t)$ and $a(t)$ state variables, the concentration in Mols of free calcium ions ($[Ca^{2+}]$) in the sarcoplasm (Fig 4C–4F), the concentration in Mols of the Calcium-Troponin complex (CaTn) in the sarcomeres, and the MU active state $a$. The interplay between the MU's NE and CE is modelled with a nonlinear dependency (dashed arrows) of the MU's Activation and Contraction Dynamics to the MU's length $\overline{l}(t)$ and activation $a(t)$, respectively. The MU's force-generating capacities also depend on some MU-specific properties (dotted lines), including its type (slow/fast) and its normalized isometric force $\overline{f_0^{MU}}$.

the whole muscle dynamics in isometric contractions were described with Eq 10. In Eq 10, at time $t$ and for a population of $n$ modelled MUs, $F^M$ is the whole muscle force in Newtons, $\overline{f_k^{MU}}$ is the normalized force developed by the $k^{th}$ MU ($k \in [\![1; n]\!]$), and $\overline{f_{k,0}^{MU}}$ is the MU-specific normalized maximum isometric force that the $k^{th}$ MU can develop. In Eq 10, the MU active states $a_k(\overline{l})$ and Force-Length (FL) scaling factors $f_{FL}(a_k, \overline{l})$ were derived from the input spike trains $sp_k(t)$ and the common MU length $\overline{l}$ with six cascading mathematical models of the MU's excitation, activation, and contraction dynamics summarized in Fig 3. Finally, because the MUs build force upon the discrete activity of their $IR_k = N_f^{tot} \cdot \overline{f_{k,0}^{MU}}$ fibres that receive the same MN action potential with a random time delay $\alpha_c$, the calculation of the normalized MU forces $\overline{f_k^{MU}}$ in Eq 10 was updated to $\overline{F_k^{MU}}$ to account for this non-simultaneous twitch activity, which acts as a low-pass filter.

$$F^M(t) = F_0^M \cdot \overline{F^M}(t)$$

$$\overline{F^M}(t) = \sum_{k=1}^{n} \overline{F_k^{MU}}(t)$$

$$\overline{F_k^{MU}}(t) = \frac{1}{IR_k} \cdot \sum_{c=0}^{IR_k} \overline{f_k^{MU}}(t + \alpha_c), \alpha_c \in \left[ -\frac{\Delta t}{2}; \frac{\Delta t}{2} \right]$$

$$\overline{f_k^{MU}}(t) = \overline{f_{k,0}^{MU}} \cdot (a_k(t, \overline{l}) \cdot f_{FL}(t, a_k, \overline{l}))$$

$$\overline{l} = 1.16 \tag{10}$$

For improved readability, the subscript $k$ was discarded in the following of the Methods section to refer to the dynamics of a representative MU of the pool.

**4.2. Contractile element and contraction dynamics—MU force-length scaling factor $f_{FL}$.** The MU $f_{FL}$ scaling factor in Eq 10 was defined with a normalized FL relationship and entirely describes the MU's contraction dynamics (Fig 3). To the authors' knowledge, no FL relationship was measured at the MU scale in humans. Despite limitations related to the multiscale approach [17], a human Force–Sarcomere Length relationship obtained from measurements in skinned human fibres [7] was here considered the most suitable option. The sarcomere-to-MU multiscale assumption is here acceptable as muscles modelled as a scaled sarcomere for their normalized isometric FL properties can provide accurate Hill-type model predictions for mammalian whole muscles [61]. Other available solutions were considered less appropriate, such as torque-length and torque-angle measurements in humans which are affected by muscle-tendon dynamics and muscle co-contraction, or FL measurements from fibre bundles in cat, rabbit and rat specimens, that have different optimal sarcomere lengths than humans [62]. Rather than a piecewise linear relationship consistent with the sliding filament theory at the sarcomere scale [63], a gaussian mathematical description in Eq 11 was fitted to the FL experimental data (red curve and black dots in Fig 4A) to account for the smoother relationship with larger plateau observed at larger scales [36,64]. For higher physiological credibility and improved prediction accuracy [65], the MU FL relationships were made nonlinearly dependent to the MU activation states $a(t)$ [20] in Eq 11 at submaximal contraction levels (dashed lines in Fig 4A).

$$f_{FL}(\bar{l}, a) = \exp\left(-\left(\frac{\bar{l} - \bar{l}_0(a)}{0.45}\right)^2\right)$$

$$\bar{l}_0(a) = 0.15(1 - a) + 1 \tag{11}$$

**4.3. MU excitation dynamics.** *Dynamics of MN AP elicitation.* The time-history of the MU active state $a(t)$ in Eq 10 was inferred by first transforming the input binary train of MN spikes $sp(t)$ into a train $e(t)$ of MN action potentials (APs) (Fig 3). The MN APs were phenomenologically modelled with Eq 12 as analytical sine waves [28] of half period $\frac{T}{2}$ and voltage amplitude $V_e$. The waves are triggered at the discharge times $t_i$ when $sp(t = t_i) = 1$, while the MN membrane remains at equilibrium, i.e., $e(t) = 0$ for $t \notin \left[t_i; t_i + \frac{T}{2}\right]$. The individual AP shapes recorded in the literature of anesthetized and dissected cats previously reviewed [17] were fitted with Eq 12, yielding $V_e = 90\,mV$ and $T = 1.4\,ms$ (Table 1). $T$ is here shorter (Fig 4B) than the amphibian $T = 2\,ms$ [66] used in a previous neuromuscular model [28].

$$\begin{cases} sp(t = t_i) = 1 \\ e(t) = V_e \sin\left[\frac{2\pi}{T}(t - t_i)\right] & \text{for } t_i \leq t \leq t_i + \frac{T}{2} \\ e(t) = 0 & \text{otherwise} \end{cases} \tag{12}$$

*Dynamics of fibre AP elicitation.* The train of the fibre APs ($u(t)$ in Fig 3) was modelled in Eq 13 as the 2nd-order response [28] to the train $e(t)$ of MN APs. In Eq 13, when the fibre receives an isolated MN discharge, $a_1$ determines the peak-to-peak amplitude $A_p$ of the fibre AP, while $a_2$ and $a_3$ respectively determine its time-to-peak duration $t_{tp}$ and exponential decay time constant $\tau_d$. The fibre APs simulated with the tuned $a_i$ coefficients in Table 1 matched

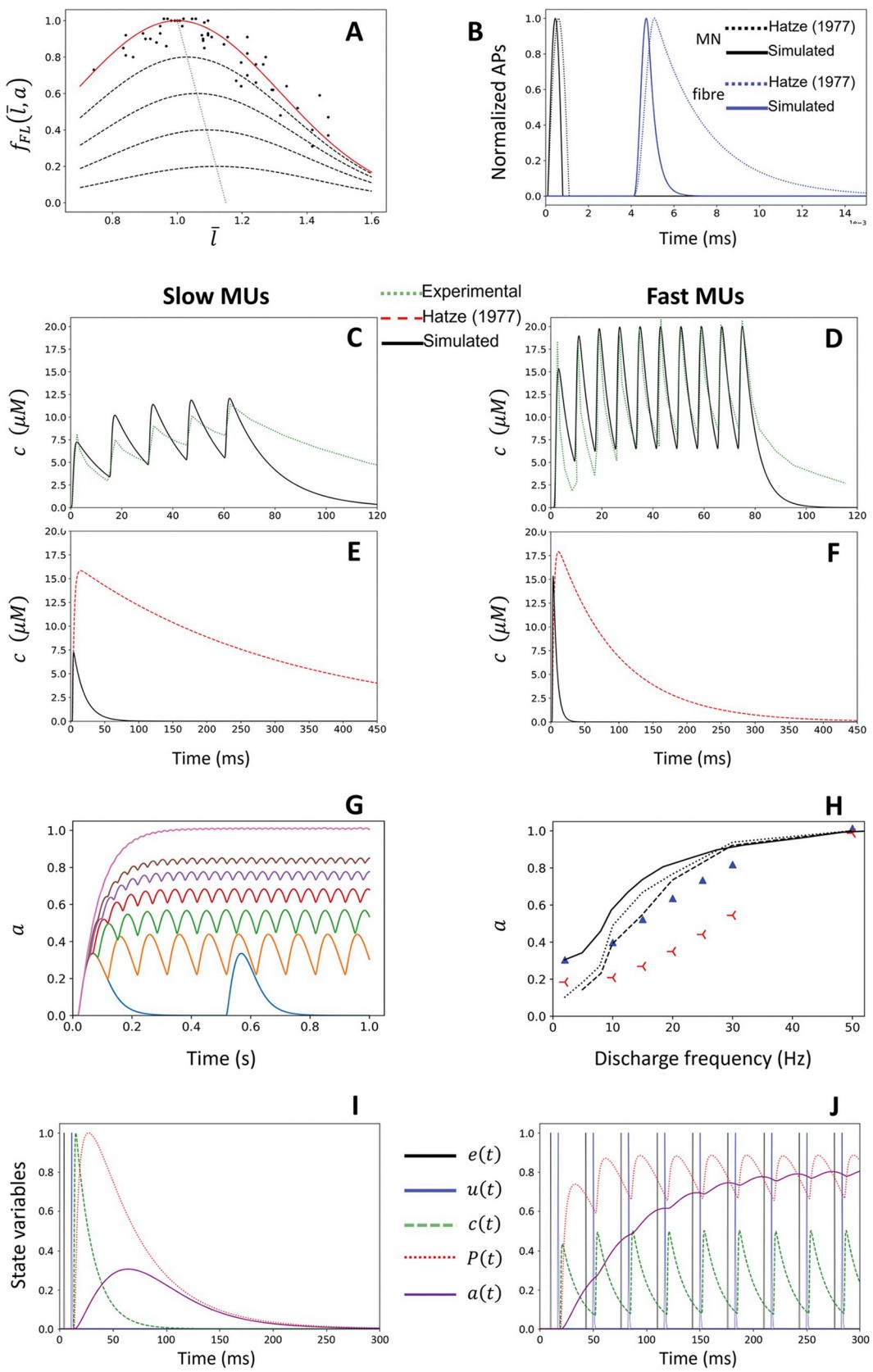

**Fig 4. Contraction, Excitation, and Activation Dynamics (Fig 3) of the slow and fast motor units (MUs) constituting the MN-driven neuromuscular model.** (A) Normalized FL relationship describing the scaling factor $f_{FL}$. The experimental data points (dots), obtained from human skinned fibres [7], were fitted (solid red curve) with Eq 11. $f_{FL}$ is nonlinearly dependent on the MU active state $a$ at submaximal levels (dashed lines) [20], where the apparent optimal MU length $l_0^M$ increases by up to 15% (dotted line). (B) Time-history of the MN (black curves) and fibre (blue curves) normalized APs simulated with Eq 12 and Eq 13, respectively. The solid lines were obtained after tuning the coefficients in those equations (Table 1) to fit cat experimental data, while the dotted lines, reported for comparison, were obtained from simulations using the original amphibian data as in Hatze's work [28]. (C-F) Time-course of the free calcium concentration $c$ in $\mu M$ for slow (C, E) and fast (D, F) MUs. The calcium transients simulated after tuning the coefficients in Eq 14 to match experimental rodent data (Table 1) are displayed with solid black lines. The simulated transients were validated against other experimental [78,79] traces (green dotted lines) at 67Hz (C) and 125Hz (D) stimulation, respectively. In (E, F), the simulated transients were compared against those proposed in Hatze's work [28] for amphibians (red dotted curves). (G, H) Relationship between the amplitude of the simulated active state $a$ and the MN firing frequency in Hz. (G): for slow-type MUs, time-history of the active state $a(t)$ with increasing MN firing frequency at 2 (blue), 10 (orange), 15 (green), 20 (red), 25 (purple), 30 (brown) and 50 Hz (pink) where the activation twitches fuse to reach 1.0 at steady-state. (H) Activation-Frequency relationship in the steady-state. Simulations for slow and fast-type MUs are displayed in blue triangles and red symbols respectively. Experimental torque-frequency relationships obtained from FDL (dotted trace, [87]), EDB (solid trace, [86]) and thenar (dashed trace, [85]) muscle fibres are superimposed. (I, J) The bottom row displays the time-course of the five state variables describing the dynamics of the Neuromechanical Element when the MU receives a unique nerve impulse (I) and a train of nerve impulses at 30Hz (J). Black and blue solid curves: trains of MN and muscle unit action potentials $e(t)$ and $u(t)$. Green dashed curve: free calcium concentration $c(t)$. Red dotted curve: CaTn concentration $P(t)$. Purple solid curve: MU active state $a(t)$. Except for the MU active state, the quantities were normalized to their maximum physiological values for visual purposes.

typical experimental cat values for $A_p$ (77 mV), $t_{tp}$ (0.6 ms) and $\tau_d$ (0.6 ms) [67–77], and were five times shorter (Fig 4B) than previously proposed for amphibians at low temperature [28].

$$\frac{d^2u}{dt^2} = a_1 e(t) - \left(a_2 u + a_3 \frac{du}{dt}\right) \tag{13}$$

**4.4. MU activation dynamics.** *Dynamics of free calcium concentration.* The MU excitation state $u(t)$ was modelled to control the level $c(t)$ of calcium concentration $[Ca^{2+}]$ in the MU sarcoplasm (Fig 3) with the 2nd-order ordinary differential equation (ODE) in Eq 14, although alternative reviewed approaches were previously proposed [17]. In Eq 14, when the fibre receives an isolated fibre AP, $b_1$ determines the peak-to-peak amplitude $A_c$ of the $[Ca^{2+}]$ twitch, and $b_2$ and $b_3$ its time-to-peak duration $t_{tp}$ and decay time constant $\tau_d$. The equation proposed in [28] was modified to introduce the length-dependent scaling factors $f_1(\bar{l})$ and $f_2(\bar{l})$, defined in Eq 15 and Eq 16, that respectively make $A_c$ and $\tau_d$ nonlinearly dependent to the normalized MU length $\bar{l}$. The procedure taken to tune the $c_i$ coefficients and derive the $f_i(\bar{l})$ scaling factors to match available experimental data for amphibian and mammal slow and fast

**Table 1. Coefficients scaling Eq 12 to Eq 18 that describe the MUs' Excitation and Activation dynamics for both slow-type and fast-type MUs.** Those coefficients were tuned to match contemporary experimental data on mammals at body temperature.

| | | slow | fast | Unit | | | slow | fast | Unit |
|---|---|---|---|---|---|---|---|---|---|
| Eq 12 | $T$ | 1.4 | | $s$ | Eq 17 | $c_1$ | $6 \cdot 10^{12}$ | $1 \cdot 10^{12}$ | $M^{-2} \cdot s^{-1}$ |
| | $V_e$ | 90 | | $V$ | | $c_2$ | 21 | 41 | $s^{-1}$ |
| Eq 13 | $a_1$ | $9 \cdot 10^7$ | | $s^{-2}$ | | $P_0$ | $1.7 \cdot 10^{-4}$ | $3.8 \cdot 10^{-4}$ | $M$ |
| | $a_2$ | $5 \cdot 10^7$ | | $s^{-2}$ | Eq 18 | $d_1$ | $1.00 \cdot 10^5$ | | $M^{-1} \cdot s^{-1}$ |
| | $a_3$ | $2 \cdot 10^4$ | | $s^{-1}$ | | $d_2$ | 0.024 | | $s^{-1}$ |
| Eq 14 | $b_1$ | 0.4 | 0.9 | $M \cdot V^{-1} \cdot s^{-2}$ | | $d_3$ | 270 | | $M^{-1} \cdot s^{-1}$ |
| | $b_2$ | $1.5 \cdot 10^5$ | $4.3 \cdot 10^5$ | $s^{-2}$ | | | | | |
| | $b_3$ | $2.5 \cdot 10^3$ | $2.4 \cdot 10^3$ | $s^{-1}$ | | | | | |

fibres *in vitro* is reported in S1 Text (Section 3).

$$\frac{d^2c}{dt^2} = b_1 \cdot u(t) - \frac{1}{f_1(\bar{l})}\left(b_2 \cdot f_2(\bar{l}) \cdot c + b_3 \cdot \frac{dc}{dt}\right) \tag{14}$$

$$\begin{cases} f_1(\bar{l}) = 0.8 & \text{if } \bar{l} \leq 1.0 \\ f_1(\bar{l}) = 0.8 + 1.33(\bar{l} - 1.0) & \text{if } \bar{l} \leq 1.15 \\ f_1(\bar{l}) = 1.0 & \text{if } \bar{l} \leq 1.30 \\ f_1(\bar{l}) = 1.0 - 0.6(\bar{l} - 1.3) & \text{if } \bar{l} > 1.30 \end{cases} \tag{15}$$

$$\begin{cases} f_2(\bar{l}) = 1.0 & \text{if } \bar{l} \leq 1.15 \\ f_2(\bar{l}) = 1.0 - 0.4 \cdot (\bar{l} - 1.15) & \text{if } \bar{l} > 1.15 \end{cases} \tag{16}$$

As reported in Fig 4D for fast fibres, the calcium transients simulated with the tuned $b_i$ coefficients (Table 1) compared well with the experimental trace at 125 Hz stimulation from [78] once steady-state was reached. The calcium twitches at steady-state returned the expected $t_{tp} = 2.0ms$, $\tau_d = 6.6ms$, and $A_c = 20\mu M$ with less than 5% difference. As Eq 14 does not account for some physiological mechanisms, such as receptor saturation and the CASQ-Traidin-RYR1 action, the typically larger $A_c$ and $\tau_d$ of the first experimental $[Ca^{2+}]$ twitch [79] were underestimated. Experimental data for slow type rodent fibres were available in [79] at 16°C, but not at 35°C body temperature. In [78], $A_c$, $t_{tp}$ and $\tau_d$ respectively increased by 26%, remained constant, and decreased by 56% when increasing the temperature from 16°C to 35°C for fast-type fibres. The $A_c$, $t_{tp}$ and $\tau_d$ coefficients reported at 16°C [79] were scaled to 35°C using the same scaling factors. As shown in Fig 4C, the simulated calcium transients at 35°C for slow fibres under 67Hz stimulation compared well with the scaled experimental data with similar $t_{tp}$, and $\tau_d$ of 19 and 44ms and an average underestimation of $A_c$ of 20%. The calcium twitches simulated at optimal fibre length with Eq 14 and the parameter values in Table 1 have ten times shorter half-widths and 2.3 times lower amplitudes (Fig 4E and 4F) than the twitches obtained with Hatze's model [28], which was calibrated on frog data at 9°C.

*Dynamics of Calcium-Troponin concentration.* The sarcomeric calcium-troponin (CaTn) binding-unbinding process was described with Eq 17 to predict from the free calcium concentration $c$ the time-course of the concentration $P$ of the Ca-Tn structures in the sarcomeres (Fig 3). The initial description [80] was extended to account for the nonlinear length-dependency of the CaTn transients with the $f_{3,4,5}(\bar{l})$ scaling factors. In Eq 17, $P_0$ is the total troponin concentration in the myoplasm, and $c_1$, $c_2$ are respectively the forward and backward reaction rates of the Ca-Tn binding-unbinding process. When triggered by a unique calcium twitch, $\frac{P_0}{f_4(\bar{l})}$ scales the peak-to-peak amplitude $A_t$ of the CaTn twitch, and $\frac{c_1}{f_3(\bar{l})}$ and $\frac{c_2}{f_5(\bar{l})}$ determine its time-to-peak $t_{tp}$ and decay constant $\tau_d$. The $P_0$, $c_1$, $c_2$ coefficients, originally calibrated to match experimental force data with a neuromuscular model [80], were here tuned to match contemporary experimental data [79,81] following a method described in S1 Text (Section 4).

$$\frac{dP}{dt} = \frac{c_1}{f_3(\bar{l})} \cdot \left(\frac{P_0}{f_4(\bar{l})} - P\right) \cdot c^2 - \frac{c_2}{f_5(\bar{l})} \cdot P \tag{17}$$

The tuned $P_0$, $c_1$, $c_2$ coefficients reported in Table 1, with which the experimental $A_t$, $t_{tp}$, and $\tau_d$ quantities could be reproduced with Eq 17 with less than 5% error at all experimentally

measured lengths and for both fibre types, are strongly consistent with the typical coefficient values reported in the cited literature $c_1 = 3.6 \cdot 10^{12} \, M^{-2} \cdot s^{-1}$, $c_2 = 100 \, s^{-1}$, and $P_0 = 2.2 \cdot 10^{-4} \, M$.

*Dynamics of MU activation.* The MU active state can be defined as the normalized instantaneous ratio of formed cross-bridges in the force-generating state in a population of myosin heads from a representative sarcomere [17]. Although semi-phenomenological models of multi-state cross-bridge attachments [82] are available, a phenomenological model [83] was used to infer the MU active state $a$ from the CaTn concentration $P$ (Fig 3) to reduce the computational cost, the number of parameters to tune, and the overall model complexity. In Eq 18, $d_1$, $d_2$ and $d_3$ respectively control the amplitude $A_a$ of the activation twitch, its time-to-peak $t_{tp}$ and its time of half-relaxation $t_{0.5}$, and were tuned to match, lacking more suitable experimental measurements in the literature, the normalized experimental twitch torque or force recorded in individual human MUs in the TA muscle [5,55,56] following a method described in S1 Text (Section 5).

$$\frac{da}{dt} = d_1 \cdot P - \frac{a}{d_2 + d_3 \cdot P} \tag{18}$$

Because of the MU type-specificity of the $[Ca^{2+}]$ and CaTn dynamics modelled previously, a unique set of values was obtained for the tuned $d_1$, $d_2$ and $d_3$ coefficients (Table 1) for both slow and fast MUs, with less than 7% error in reproducing the experimental $A_t$, $t_{tp}$ and $t_{0.5}$ quantities when simulating single activation twitches for both slow and fast MUs.

When receiving artificial excitatory spike trains at 2, 10, 15, 20, 25, 30, and 50 Hz, the NE of a slow-type MU at optimal length $\bar{l} = 1$ produced the activation traces in Fig 4G. As supported by previous findings in mammals [2], the simulated activation twitches fused for MN firing frequencies above 2Hz and the initial rate of increase of the MU active state was independent from discharge frequency, while the 0-to-0.8 rate of increase increased with higher firing frequencies from 3.5/s at 15Hz and 5.8/s at 50Hz. Consistent with previous findings [84], every new elicited activation twitch increased the MU activation level at steady-state by 100% at 5Hz, and only by 1% at 50Hz, following a negative exponential tendency with increasing MN discharge frequency.

Fig 4H displays the simulated steady-state activation-frequency relationship for both slow-type MUs (blue triangles) and fast-type MUs (red symbols). Despite obvious limitations but lacking experimental measurements of human activation states in the literature, those results were compared to the literature of experimental human muscle force and torque twitches. In Fig 4H, simulated twitch-tetanus ratios of 0.30 and 0.19 were obtained at steady-state for the simulated slow and fast-type MU active states, respectively. These ratios are of the same order of magnitude as those experimentally obtained for human hand muscles (range 0.1–0.3, [85–87]). The activation-frequency results for the simulated slow-type MUs compared relatively well with the findings from these experimental studies with a maximum difference of 30% at 15Hz with the extensor digitorum brevis (EDB) muscle data [86] and less than 10% at all physiological frequencies above 5Hz with the data obtained from thenar muscles [85]. Yet, the simulated rate of increase in $a$ with firing frequency largely underestimated the available experimental data for the fast-type MUs with up to 200% difference. The lack of available experimental human data of fast MU twitches prevents for now from deriving more advanced conclusions on the accuracy of the simulated MU active states.

**4.5. Solving the dynamics of the MU pool.** Working at fixed MU length $\bar{l} = 1.16$ for each of the $n$ modelled MUs, the cascading ODEs in Eq 12 to Eq 18, that describe the MU-specific excitation and activation dynamics, were first numerically solved at each time step, yielding a vector of MU active state time-histories $a_i(t)$. Fig 4I and 4J displays the time-histories of

the five state variables that describe the NE's activity at 1 and 30 Hz, respectively. An estimation of the MU normalized forces and of the whole muscle force was then obtained with Eq 10.

## 5. Neural control to the MN-driven neuromuscular model

**5.1. Neural controls to the neuromuscular model.** In this study, we assessed the MN-driven neuromuscular model (Fig 1F) with two different types of neural controls, i.e., the experimental samples of $N_r$ identified spike trains (Fig 1C) and the complete population of $N$ MU spike trains (Fig 1D) obtained with a MU pool reconstruction method [41]. Both approaches were tested with the sets of spike trains identified with the three electrode densities (Fig 1B) at both contraction intensities up to 30% and 50% MVC.

When directly inputting the $N_r$ experimental spike trains to the neuromuscular model, the $N_r$ MUs were assigned with Eq 19 representative maximum isometric forces $\overline{f_{k,0}^{MU}}$, necessary parameter in Eq 10, to account for the force-generating capacities of their neighbouring MUs, in the sense of their recruitment thresholds, that were not identified experimentally. The muscle force $F_{N_r,1}^M(t)$ (blue dotted line in Fig 1G) was first predicted when the $N_r$ identified MUs were assumed to be homogeneously spread across the $T^{th}$-ranked MU pool, in which case the $f_0^{MU}(i)$ distribution in Eq 8 was evenly split in $N_r$ domains along the x-axis (Fig 2B). A second TA force $F_{N_r,2}^M(t)$ (green dashed line in Fig 1G) was predicted after assigning to the $N_r$ MUs representative maximum forces with Eq 19 after a physiological mapping of these MUs to the complete $T^{th}$-ranked MU pool using Eq 6. For example, when $N_r = 30$ MUs are identified at 30% MVC, where $N_a = 300$ MUs are supposed to be recruited (Eq 5), the $9^{th}$, $10^{th}$, and $11^{th}$ identified MUs in the $T^{th}$-ranked experimental sample are mapped into the complete $T^{th}$-ranked pool to the $90^{th}$, $100^{th}$, and $110^{th}$ MUs with the first method (considering even steps of $\lfloor \frac{N_a}{N_r} \rfloor = \lfloor \frac{300}{30} \rfloor = 10$ MUs) and to the $90^{th}$, $120^{th}$, and $180^{th}$ MUs with the second method (using Eq 6). The $10^{th}$ identified MU represents 10 MUs and produces the combined force $f_{0,10}^{MU} = \sum_{j=96}^{105} f_0^{MU}(j) = 13$ N with the first method, while it represents 45 surrounding MUs and generates $f_{0,10}^{MU} = \sum_{j=106}^{150} f_0^{MU}(j) = 71$ N with the second method.

$$i \rightarrow N_i = \left\{ i \cdot \lfloor \frac{N_a}{N_r} \rfloor; \text{ or from MU mapping} \right\}, i \in [\![1; N_r]\!], N_i \in [\![1; N]\!]$$

$$N_{i,1} = \lfloor \frac{N_{i-1} + N_i}{2} \rfloor + 1$$

$$N_{i,2} = \lfloor \frac{N_i + N_{i+1}}{2} \rfloor$$

$$f_{0,i}^{MU} = \sum_{j=N_{i,1}}^{N_{i,2}} f_0^{MU}(j) \tag{19}$$

The TA force $F_N^M(t)$ (red solid line in Fig 1G) was predicted from a comprehensive cohort of $N = 400$ simulated spike trains, that described the discharge activity of a completely reconstructed MU population (red solid line in Fig 1E). The complete population of $N$ MUs was obtained from a validated computational method [41] that maps with Eq 6 the $N_r$ MUs into the MU pool and infers from the $N_r$ identified spike trains a continuous distribution of the MN electrophysiological properties across the MU pool to physiologically scale a cohort of $N = 400$ models of motoneurons that predicts the discharge activity of the $N-N_r$ MUs that

were not identified experimentally. The implementation of the method is publicly available at https://github.com/ArnaultCAILLET/Caillet-et-al-2022-PLOS_Comput_Biol. Here, the distribution of MU maximum forces $f_0^{MU}(i)$ in Eq 8 was directly applied to the reconstructed population of $N$ MUs.

**5.2. Neural drive to muscle and quality of the experimental neural control.** Cumulative spike trains were computed as the temporal binary summation of the pools of spike trains and were low-pass filtered in the bandwidth [0–4] Hz relevant for force generation [88] to approximate the effective neural drive to muscle $D(t)$. As suggested for isometric contractions [33], the normalized effective neural drive $\overline{D}(t)$ was compared for validation against the normalized experimental TA force $\overline{F_{TA}}(t)$ (black trace in Fig 1E) with calculation of three metrics: the error in seconds in identifying the onset of the neural drive $\Delta_1$, the normalized root-mean-square error (nRMSE) and the coefficient of determination $r^2$. Of note, $\overline{F_{TA}}(t)$ was obtained by normalization to the average of the $F_{TA}(t)$ values over the plateau of contraction and not to the maximum registered $F_{TA}$ value to minimize the impact of possible discharge artefacts onto the interpretation of the nRMSE metric. To discuss the importance of the quality of the experimental data in deriving a reliable neural control to the MN-driven neuromuscular model developed in this study, the validation between the $\overline{D}(t)$ and $\overline{F_{TA}}(t)$ traces was performed for both the pools of $N_r$ experimental and $N$ reconstructed spike trains (blue dotted and plain red traces in Fig 1E, respectively) derived from the three grids of increasing electrode density (Fig 1B).

## 6. Validation of the predicted forces

The TA forces $F^M(t)$ predicted with the MN-driven neuromuscular model (Fig 1F) were validated against the experimental TA force $F_{TA}(t)$ (solid black trace in Fig 1G), that was estimated from the recorded ankle torques $T(t)$ with Eq 4. The $F^M(t)$ versus $F_{TA}(t)$ validation was performed with calculation of {1} the error in seconds in identifying the onset $\Delta_1$ of generated force, i.e. when 2% of the max generated force was produced, {2} the experimental force $\Delta_1^F$ developed after the $\Delta_1$ delay, {3} the maximum error (*ME*) in Newtons, {4} the nRMSE in percentage, and {5} $r^2$. The nRMSE was calculated for the complete contraction, the ascending ramp ($nRMSE_{r1}$), the plateau ($nRMSE_p$), and the descending ramp ($nRMSE_{r2}$). $\Delta_1$ and $\Delta_1^F$ assess the delay in predicting the onset of force and only depends on the first discharge time identified experimentally [41], *ME* identifies the maximum error in predicting the whole muscle force and is related to the $\Delta_1$ and $\Delta_1^F$ metrics in the regions of low generated force, the nRMSE metric evaluates the accuracy in predicting the amplitude of the predicted whole muscle force, which mainly relies on both the distribution of MU tetanic forces and the accurate prediction of the MU active states, and $r^2$ evaluates the time-course of the variation of the muscle force, which mainly relies on the distribution of the MU neuromechanical properties across the MU pool.

## Results

### 1. Subject-specific MSK quantities and experimental TA force

The subject-specific MSK model (Fig 1I) yielded the subject-muscle-specific properties in Table 2 for the TA muscle and for its agonist EDL and EHL, and antagonist muscles SOL, GM, and GL in ankle dorsiflexion. By reapplying with Eq 4 the exponential $\Delta T(T(t))$ relationship derived from the bEMG and torque recordings obtained during the second experimental session to the torque traces $T(t)$ recorded during the first experimental session, the experimental TA force $F_{TA}(t)$ (solid red traces in Fig 5) was estimated (see S1 Text (Section 1) for details of the calculation). In the normalized space (bottom row in Fig 5), $\overline{F_{TA}}(t)$ compared well to the

**Table 2. Summary of the subject-muscle-specific MSK properties of the six flexor muscles when the ankle joint is set to a 30° plantarflexion angle.** The moment arm $L$ made by the tendons with the ankle joint and the muscle-tendon lengths $l^{MT}$ were identified by MRI segmentation and measurements in NMSBuilder [52] and OpenSim [48,49]. The optimal length $l_0^M$ and the tendon slack length $l_s^T$ were scaled from the generic values reported in [54] with the ratio of generic to subject-specific muscle-tendon lengths $l^{MT}$ at default 0° plantarflexion angle. The muscle maximum isometric force $F_0^M$ was estimated from the muscle volume $V^M$, which was obtained from MRI segmentation for the TA, and from anatomical relationships [47] for the other muscles. The normalized muscle lengths $\bar{l}$, at 30° angle in ankle plantarflexion and assuming a rigid tendon, were calculated with Eq S1 in S1 Text (Section 1).

| Muscle | $L[mm]$ | $F_0^M[N]$ | $l^{MT}[mm]$ | $l_0^M[mm]$ | $l_s^T[mm]$ | $\bar{l}$ |
|--------|---------|------------|--------------|-------------|-------------|-----------|
| TA | 25.5 | 1046 | 319 | 68.2 | 240 | 1.16 |
| EDL | 26.1 | 403 | 525 | 80.5 | 462 | 0.78 |
| EHL | 25.9 | 176 | 487 | 87.6 | 382 | 1.20 |
| SOL | 51.1 | 3822 | 373 | 54.1 | 340 | 0.60 |
| GM | 51.8 | 2232 | 447 | 52.6 | 411 | 0.68 |
| GL | 52.2 | 1058 | 448 | 62.4 | 398 | 0.80 |

muscle force that would be computed if the TA alone produced the total recorded ankle torque $T$ (black dashed traces in Fig 5), with a maximum error $\leq 15\%$, nRMSE $\leq 7\%$, and $r^2 \geq 0.98$. Yet, ignoring the co-contracting activities of the EDL, EHL, SOL, GM, and GL muscles would

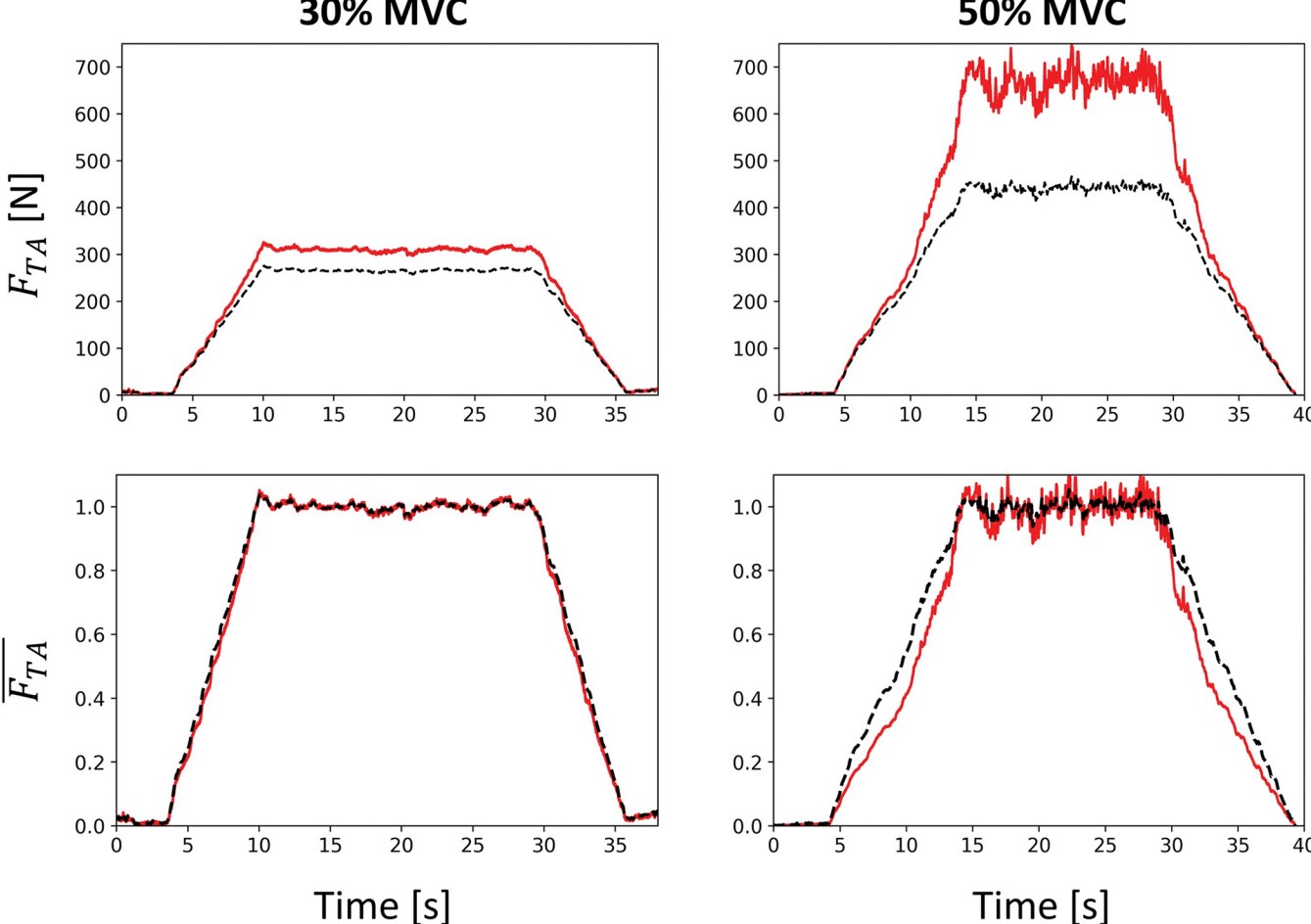

**Fig 5.** Estimation of the force developed by the TA muscle during the trapezoidal dorsiflexions up to 30% (left) and 50% MVC (right) performed during the first experimental session. The TA forces (first row: in Newtons, second row: normalized), displayed in solid red traces, were estimated by correcting the recorded torque $T$ for co-contraction with the relationship in Eq 4 (See S1 Text (Section 1) for details about the calculation). The black dashed traces display the TA force calculated without considering the co-contraction of surrounding muscles (as if the TA alone produced the entire recorded ankle torque $T$).

underestimate the TA force $F_{TA}$ by up to 49 N and 283 N, i.e., 18% and 60% of the maximum $F_{TA}$ value, at 30% and 50% MVC respectively (upper row in Fig 5).

## 2. Experimental neural control

For the contractions up to 30% and 50% MVC, 81, 32, and 14 MUs, and 55, 15, and 3 MUs (Fig 1C) were respectively identified [40] with the EMG grids of 4, 8, and 12 mm interelectrode distance (Fig 1B). The dataset of three identified MUs was disregarded in the following results because too few and only high-threshold MUs were identified in this dataset, which therefore provided a highly inaccurate description of the full spectrum of MU properties and discharge activity of the real MU pool [41]. The histogram distribution of the identified MUs across the MU pool according to their recruitment threshold $T^{th}$ in %MVC is displayed in Fig 6. Importantly, the denser the EMG grid and the lower the generated force, the more homogeneous the distribution of the identified MUs across the MU pool [40]. For example, 12% to 20% of the MUs decoded with the densest grid at 30% MVC were systematically identified in each of the 5% ranges sampling the recruitment range (histogram in Fig 6C). Higher heterogeneity and lower representation of low-threshold MUs were obtained with shallower grids and at 50% MVC. Classically, the decomposition algorithms converge towards the large MUs that

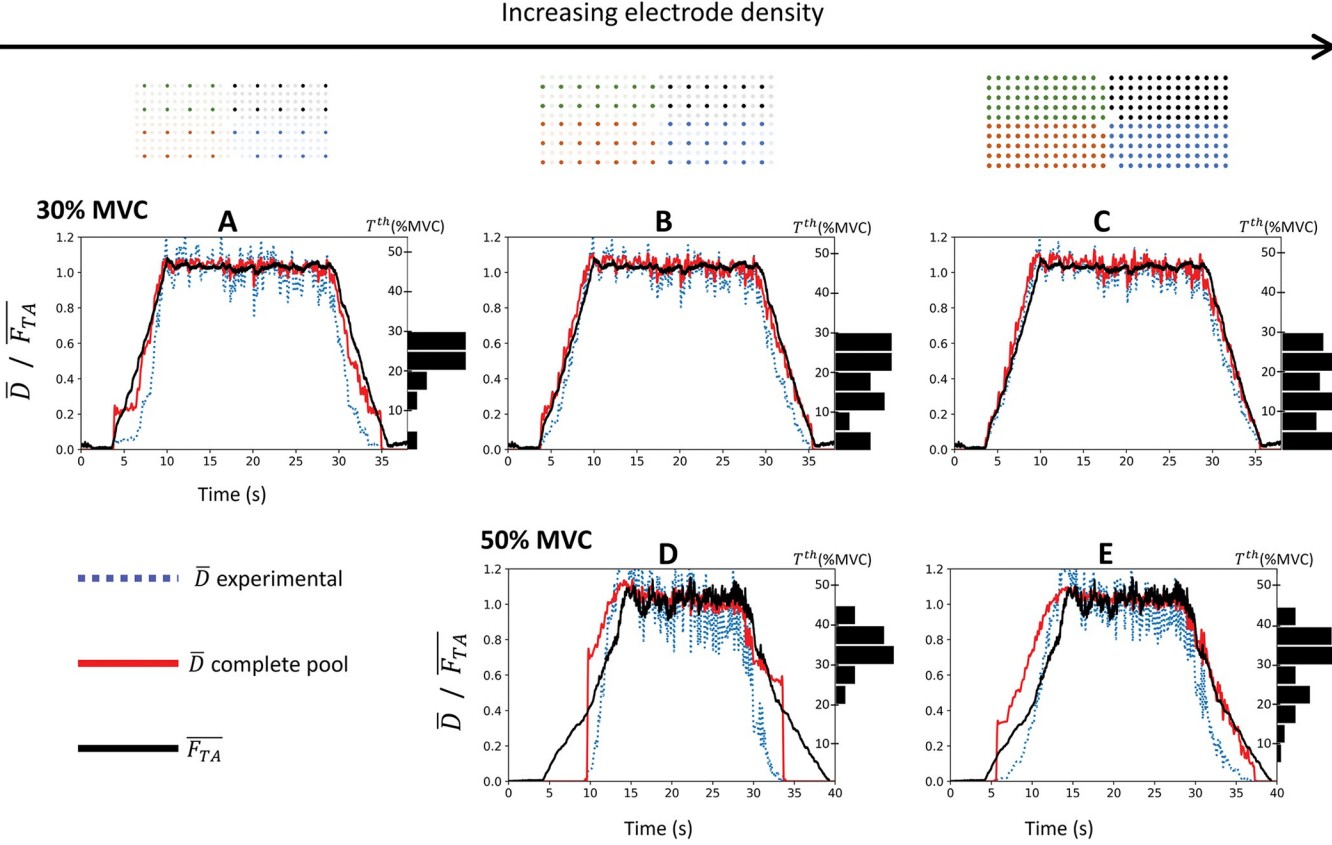

**Fig 6.** For the three grid configurations of increasing electrode density (in columns) and for both contractions up to 30% MVC (first row: A, B, C) and 50% MVC (second row: D, E), validation against the experimental normalized force trace $\overline{F_{TA}}(t)$ (solid dark trace) of the normalized effective neural drive to muscle $\overline{D}(t)$ estimated from the completely reconstructed population of $N = 400$ MUs (solid red traces) and from the experimental samples of $N_r$ identified MUs (dotted blue traces). The TA force $F_{TA}(t)$ (Fig 5) was estimated from the recorded ankle torque $T$ with Eq 4. The histogram distributions of the identified MUs across the recruitment range $T^{th}$ in %MVC were normalized to the number of MUs identified experimentally.

contribute most to the EMG signals, while action potentials of the small MUs of lower energy are masked by the potential of larger units, more of which are recruited at 50% MVC than at 30% MVC. At 30% MVC, increasing the electrode density better samples the action potential profiles of the low-threshold unmasked MUs across multiple electrodes, enabling their identification. The normalized effective neural drive to muscle $\overline{D}(t)$ (Fig 1E) was computed with these cohorts of spike trains and was validated against the experimental normalized TA force $\overline{F_{TA}}(t)$ (black solid traces in Fig 6) with the three validation metrics reported in Table 3.

The normalized effective neural drive $\overline{D}(t)$ estimated from the experimental samples of $N_r$ MUs (blue dotted lines in Fig 6) was always better approximated when the electrode density increased, especially in the regions of low generated forces. For example, at 30% MVC, it was obtained $\Delta_1 = 0.07$ $s$, $nRMSE = 8.0\%$, and $r^2 = 0.97$ from the 81 discharging MUs identified with 4 mm interelectrode distance, while the 14 MUs identified with 12 mm interelectrode distance returned $\Delta_1 = 0.14$ $s$, $nRMSE = 17.5\%$, and $r^2 = 0.89$ (Table 3). The effective neural drive $\overline{D}(t)$ estimated from the experimental MU samples was always better approximated at 30% MVC than at 50% MVC (blue dotted traces in upper row versus lower row in Fig 6), with two to three times lower nRMSE values, and strongly lower $\Delta_1$ and higher $r^2$ values, respectively (Table 3).

Complete populations of $N = 400$ MNs (Fig 1D) were also reconstructed from the discharge activity of the $N_r$ identified MNs by estimating the discharge activity of the MUs not identified experimentally [41]. In this case, the $\overline{D}(t)$ estimated at 30% MVC (solid red lines, upper row, in Fig 6) was accurately predicted and was unrelated to the quality of the input experimental samples, with $nRMSE < 7\%$, and $r^2 = 0.98$ for the three EMG grid configurations (Table 3). At 50% MVC, the $\overline{D}(t)$ derived from the $N$ MNs (solid red lines, bottom row, in Fig 6) was better estimated when the MU pool was reconstructed from denser grids of electrodes, with lower nRMSE and higher $r^2$ values (Table 3). As for the experimental MU samples, $\overline{D}(t)$ estimated from the $N = 400$ MUs was better approximated at 30% MVC than at 50% MVC (solid red traces in upper row versus lower row in Fig 6). At 50% MVC, the reconstructed populations of discharging MUs returned inaccurate estimations of the onset and ascending ramp of force, with $\Delta_1 > 1s$, $nRMSE > 12\%$, and $r^2 \leq 0.95$. Overall, the normalized effective neural drive $\overline{D}(t)$ was systematically better approximated by the reconstructed pool of $N = 400$ MUs than by the experimental samples of $N_r$ identified MUs (solid red versus blue dotted traces in Fig 6) with lower nRMSE and higher $r^2$ values (Table 3).

Therefore, $\overline{D}(t)$ was best approximated by samples of MU spike trains of identified MUs that spanned across the whole recruitment range and followed a homogeneous distribution across the MU pool. Obtaining such representative description of the discharge behaviour of the real MU pool was possible with EMG signals recorded at 30% MVC with high electrode

**Table 3. Validation of the normalized effective neural drive to muscle $\overline{D}(t)$ for both contractions up to 30% and 50% MVC, computed from the spike trains obtained with the three grid configurations of 4, 8, and 12 mm interelectrode distance that involve 256, 64, and 36 EMG electrodes respectively.** The validation was performed for both the experimental samples of $N_r$ identified MUs, and for the completely reconstructed pool of $N$ MUs.

| Contraction | Electrode density (mm) | $\Delta_1(s)$ | $N_r$ MNs (experimental) | | $N$ MNs (simulated) | |
|---|---|---|---|---|---|---|
| | | | nRMSE (%) | $r^2$ | nRMSE (%) | $r^2$ |
| 30% MVC | 4 | 0.1 | 8.0 | 0.97 | 5.3 | 0.98 |
| | 8 | 0.1 | 10.5 | 0.95 | 5.3 | 0.98 |
| | 12 | 0.1 | 17.5 | 0.89 | 6.3 | 0.90 |
| 50% MVC | 4 | 1.2 | 16.4 | 0.87 | 12.9 | 0.95 |
| | 8 | 5.2 | 24.2 | 0.77 | 16.4 | 0.89 |

density (4 mm interelectrode distance) or, to some extent, with the computational reconstruction of the complete MU pool, both of which aimed to correct for the systematic bias of HDEMG decomposition towards identifying the higher-threshold MUs (histograms in Fig 6) of the discharging MU pool [40].

## Activation and Contraction dynamics of the MU pool and TA Force validation

The neuromuscular model was built as populations of $N_r$ or $N$ in-parallel FGs, whether it received as neural control the experimental samples of $N_r$ identified spike trains (Fig 1C) or the completely reconstructed populations of $N$ MU spike trains (Fig 1D). According to Fig 2, around 298 and 355 MUs were recruited over the trapezoidal contractions up to 30% and 50% MVC, respectively, in which case all the modelled discharging MUs were assigned slow-type properties for their activation dynamics (Fig 3).

Fig 7A displays the distribution of the maximum values of the modelled MUs' activation states over the contraction up to 30% MVC, that were estimated from the cohorts of $N_r = 81$ (green triangles) and $N = 400$ (red dots) spike trains with Eq 12 to Eq 18. Consistent with the onion skin theory [89] where MN discharge rate and recruitment threshold are negatively related, the estimated MU activation was overall negatively related to recruitment threshold, with low-threshold MUs reaching maximum activation states up to $a = 0.80$ and higher-threshold MUs as low as $a = 0.18$. Because of trendline fitting during the reconstruction process of the MU pool [41], the monotonically decreasing distribution of active states of the $N$-population did not account for the physiological variability in MN discharge rate with recruitment threshold (green triangles in Fig 7A). The same conclusions were obtained with the neural controls obtained with the grids of lower EMG electrode densities and at 50% MVC.

Fig 7B–7D displays the distribution of the maximum forces (blue crosses, green triangles, red dots) developed by the modelled MUs over the contraction up to 30% MVC, that were estimated with Eq 10 from the MUs' active states $a(t)$, the common MU length $\bar{l}$, and the MU-specific maximum isometric forces $f_{k,0}^{MU}$ (black dots), for cohorts of $N_r = 81$ (Fig 7B and 7C) and $N = 400$ (Fig 7D) MUs. Depending on the modelling approach, the modelled MUs were assigned maximum MU forces $f_{k,0}^{MU}$ (black dots in Fig 7B–7D) by either blindly assuming the $N_r$ MUs to be evenly distributed across the MU pool (Fig 7B), mapping the $N_r$ MUs to the real MU pool with Eq 6 (Fig 7C), or directly applying the $f_0^{MU}(k)$ distribution in Eq 8 to the complete MU pool (Fig 7D). Depending on the approach, the individual MUs produced maximum forces $f_k^{MU}$ over the contraction up to 30% MVC in the range 1.8–6.6 N, 0.4–16.5 N, and 0.5–1.5N (blue crosses in Fig 7B, green triangles in Fig 7C, and red dots in Fig 7D, respectively), and in the range 4.2–16.2 N, 1.5–136.7 N, and 0.7–2.2 N at 50% MVC. Only the reconstructed pool of $N = 400$ MUs (Fig 7D) provided a window onto the continuous distribution of the dynamics of the MU pool with physiological generated MU forces.

The simulated MU forces were processed and linearly summed with Eq 10 to yield the three predicted whole muscle forces in Fig 8A–8E $F_{N_r,1}^M(t)$ (experimental MU population with blind $f_{k,0}^{MU}$ distribution—blue dotted traces), $F_{N_r,2}^M(t)$ (experimental MU population with MU mapping—green dashed traces), and $F_N^M(t)$ (reconstructed MU population—solid red traces), which were validated against the experimental TA force $F_{TA}(t)$ (solid black traces) with the validation metrics in Table 4.

When high numbers of MUs were homogeneously identified over the full range or recruitment ($N_r = 81$ MUs, 30% MVC, 4 mm interelectrode distance), i.e., when $\bar{D}(t)$ was accurately estimated from the experimental samples of spike trains (Fig 6C), $F_{TA}$ was predicted by

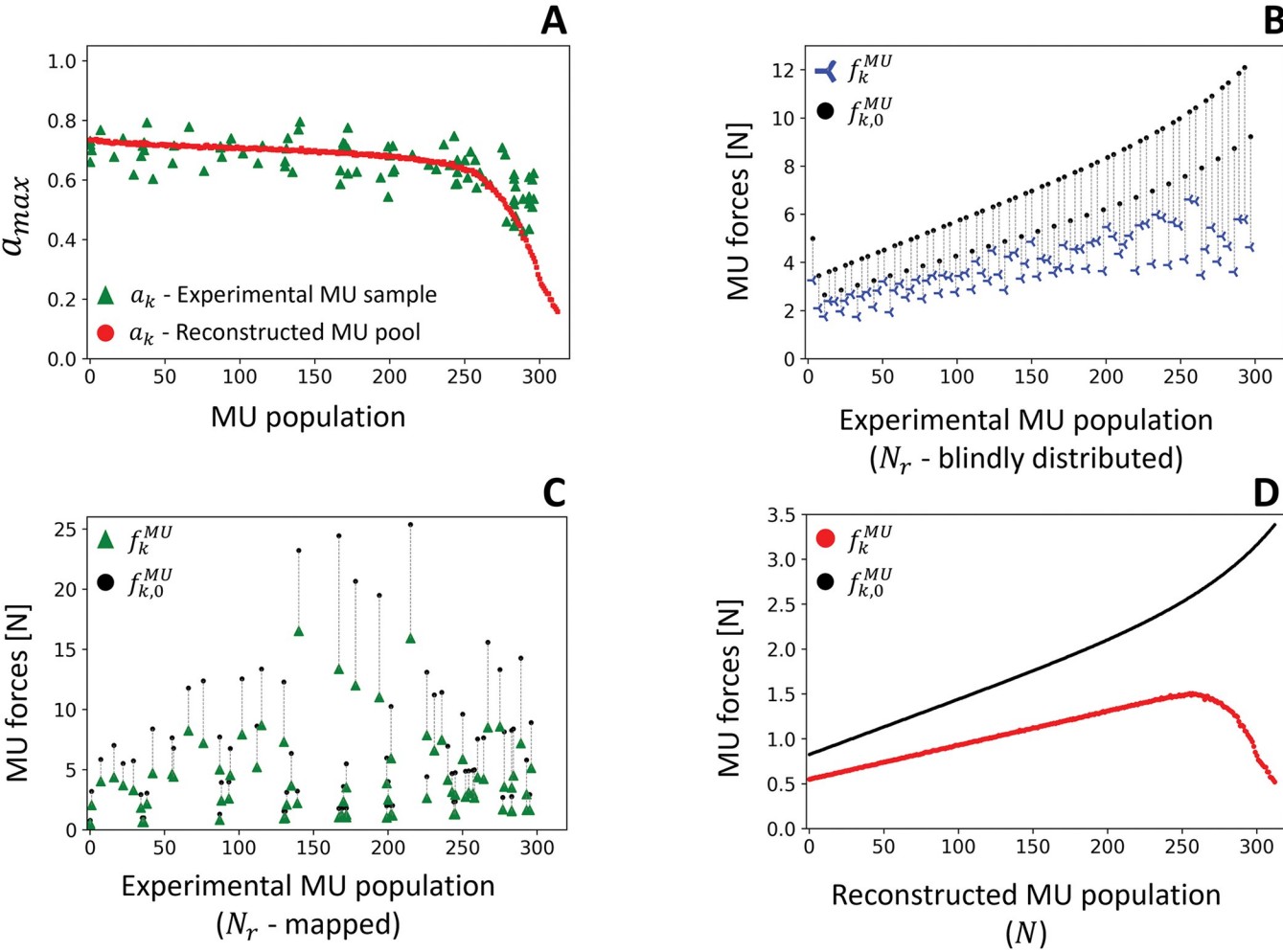

**Fig 7.** Distribution across the $T^{th}$-ranked MU population of the maximum values at 30% MVC of the MU activation states $a_k$ (A) and MU forces $f_k^{MU}$ (B-D) simulated with the MN-driven neuromuscular model. The model either received as neural input the $N_r = 81$ experimental spike trains (green triangles in A for $a_k$; plots B and C for $f_k^{MU}$) or the discharge activity of the complete pool of $N = 400$ MUs (red dots in A for $a_k$; plot D for $f_k^{MU}$). In B-D, the black dots represent the distribution of the MUs' maximum forces $f_{k,0}^{MU}$, which is continuous for the completely reconstructed pool (D). In B and C, the $N_r$ identified MUs were assigned representative maximum forces $f_{k,0}^{MU}$ (black dots) to account for the force-generating properties of the MUs not identified experimentally. To do so, in B, the $N_r$ identified MUs were assumed to be homogeneously distributed across the $T^{th}$-ranked MU pool, while they were accurately mapped to the real pool with Eq 6 in C. In B, the non-continuous distribution of the $f_{k,0}^{MU}$ values is explained by the fact that $\frac{N}{N_r}$ is not an integer value in Eq 19.

$F_{N_r,1}^M$, $F_{N_r,2}^M$, and $F_N^M$ with the same level of accuracy (Fig 8C) for the eight validation metrics in Table 4 with $r^2 \geq 0.98$, all *nRMSE* values below 10%, and ME in 36–66%, irrespective from the three choices of $f_{0,k}^{MU}$ assignment (black dots in Fig 7B–7D).

In all other cases, $F_{TA}(t)$ was the most accurately predicted by both $F_N^M(t)$ when the complete MU pool was modelled (solid red traces in Fig 8A–8E, *nRMSE* in range 8–13% and $r^2$ in range 0.97–0.99 at 30% MVC, and *nRMSE* in range 15–16% and $r^2$ in range 0.90–0.95 at 50% MVC) and $F_{N_r,2}^M(t)$ when the population sample of $N_r$ MUs was assigned representative $f_{0,k}^{MU}$ values according to a MU mapping (green dashed traces in Fig 8A–8E, *nRMSE* in 6–11% and $r^2$ in 0.97–0.99 at 30% MVC, and *nRMSE* in 14–20% and $r^2$ in 0.92–0.95 at 50% MVC). The least accurate predictions were systematically obtained with $F_{N_r,1}^M(t)$ when the population sample of $N_r$ MUs received experimental neural inputs and was blindly assigned $f_{0,k}^{MU}$ values (blue

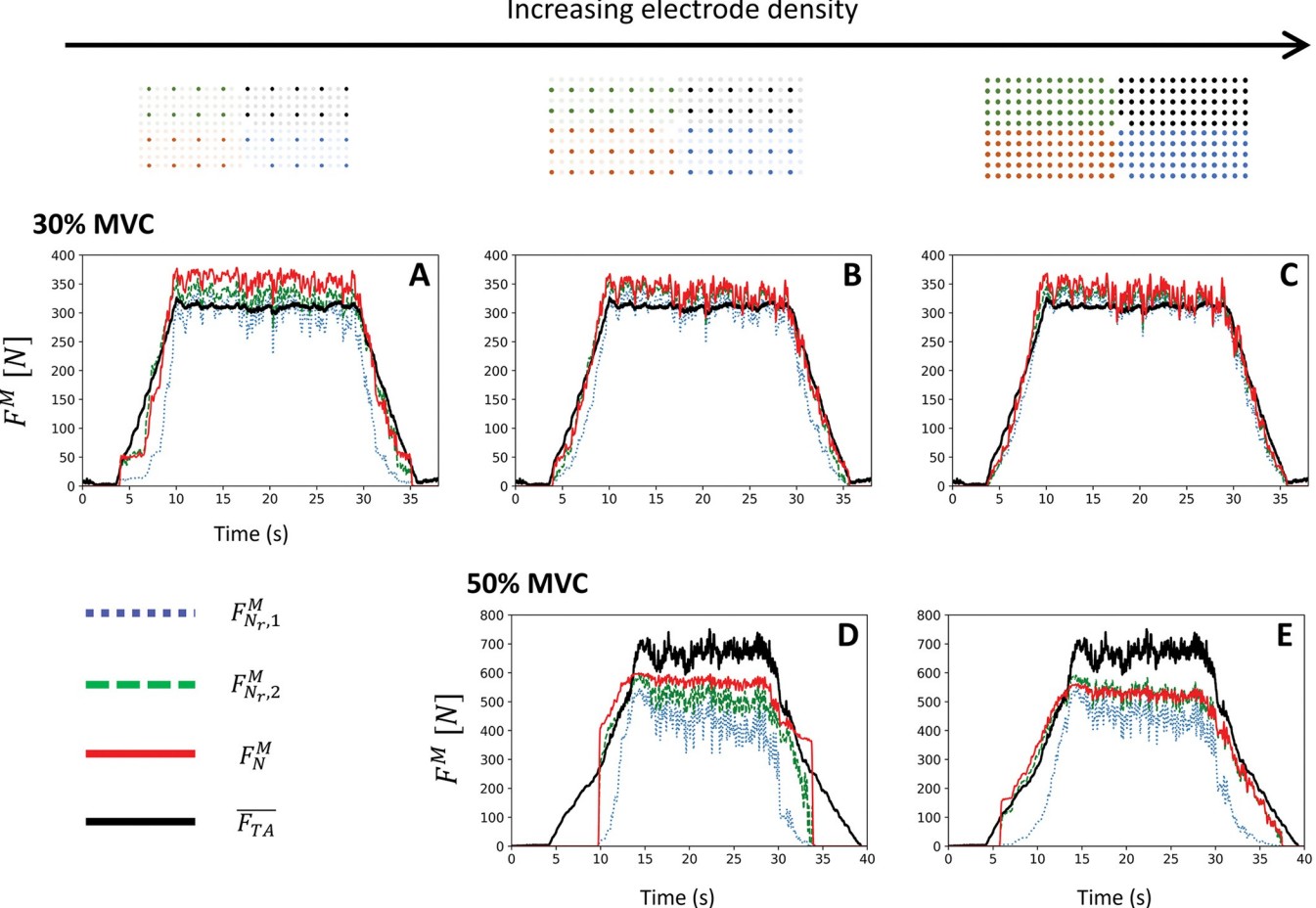

**Fig 8.** For the three grid configurations of increasing electrode density (in columns) and for both contractions up to 30% MVC (first row: A, B, C) and 50% MVC (second row: D, E), validation against the experimental force trace $F_{TA}(t)$ (solid dark trace) of the TA force $F^M(t)$ estimated with the MN-driven neuromuscular model in Fig 1F. $F^M(t)$ was estimated from the completely reconstructed population of $N = 400$ MUs ($F_N^M(t)$, solid red traces) and from the experimental samples of $N_r$ identified MUs when 'blind' ($F_{N_r,1}^M(t)$, blue dotted traces) and $T^{th}$-informed ($F_{N_r,2}^M(t)$, green dashed traces) distributions of the MU maximum isometric forces $f_0^{MU}(i)$ were assigned to the MU samples. The experimental TA force $F_{TA}(t)$ was estimated from the recorded ankle torque $T$ with Eq 4.

dotted traces in Fig 8A–8E, *nRMSE* in 7–18% and $r^2$ in 0.89–0.98 at 30% MVC, and *nRMSE* in 27–32% and $r^2$ in 0.86–0.92 at 50% MVC). $F_N^M(t)$ and $F_{N_r,2}^M(t)$ performed better than by $F_{N_r,1}^M(t)$ especially over the ramps of force with around 1.5–3 times lower *ME*, $nRMSE_{r1}$, and $nRMSE_{r2}$ values ($nRMSE_{r1}$ and $nRMSE_{r2}$ metrics calculated over the ramps of force—Table 4), because the blind distribution of $f_0^{MU}$ values in $F_{N_r,1}^M$ (Fig 7B) did not correct, contrary to the two other approaches, for the experimental bias towards identifying relatively more high-threshold MUs with decomposed HDEMG signals and overestimated their force-generating activities in the MU pool. As long as the $N_r$ decoded MUs were homogeneously spread across the identified portion of the MU pool, $F_{N_r,2}^M$ returned slightly more accurate predictions of $F_{TA}$ than $F_N^M$, with slightly lower *nRMSE* and maximum error values over the whole contraction. Mainly, $F_N^M$ returned higher $nRMSE_p$ values (Table 4) calculated over the plateau of force because of the noisy force-generating activity of the highest-threshold recruited MUs modelled with the reconstruction method [41]. With experimental samples of lesser quality (50% MVC, 8 mm interelectrode distance, Fig 6), $F_N^M$ performed better than $F_{N_r,2}^M$.

**Table 4. Validation against the experimental force trace $F_{TA}(t)$ of the predicted TA force $F^M(t)$ at 30% and 50% MVC.** $F^M(t)$ was predicted from the spike trains obtained at 4-, 8-, and 12-mm electrode density. $F^M(t)$ was estimated from the completely reconstructed population of $N = 400$ MUs ($F_N^M$) and from the experimental samples of $N_r$ identified MUs when 'blind' ($F_{N_r,1}^M$) and $T^{th}$-informed ($F_{N_r,2}^M$) distributions of the MU maximum isometric forces $f_0^{MU}(i)$ were assigned to the MU samples. nRMSE, $nRMSE_{r1}$, $nRMSE_p$, $nRMSE_{r2}$ were calculated for the whole contraction, and over the ascending ramp, plateau, and descending ramp, respectively.

| Contraction | Electrode density (mm) | Validated Forces $F^M(t)$ | $\Delta_1$ (s) | $\Delta_1^F$(N) | ME (N) | nRMSE (%) | $nRMSE_{r1}$ (%) | $nRMSE_p$ (%) | $nRMSE_{r2}$ (%) | $r^2$ |
|---|---|---|---|---|---|---|---|---|---|---|
| 30% MVC | **4** | $F_{N_r,1}^M$ | 0.1 | 11 | 71 | 7 | 8 | 5 | 10 | 0.98 |
| | | $F_{N_r,2}^M$ | 0.2 | 18 | 47 | 6 | 6 | 7 | 6 | 0.99 |
| | | $F_N^M$ | 0.1 | 14 | 74 | 8 | 6 | 10 | 4 | 0.99 |
| | **8** | $F_{N_r,1}^M$ | 0.3 | 28 | 94 | 11 | 14 | 5 | 16 | 0.96 |
| | | $F_{N_r,2}^M$ | 0.3 | 22 | 51 | 7 | 6 | 8 | 6 | 0.99 |
| | | $F_N^M$ | 0.3 | 28 | 57 | 8 | 5 | 10 | 4 | 0.99 |
| | **12** | $F_{N_r,1}^M$ | 0.3 | 28 | 158 | 18 | 26 | 6 | 25 | 0.89 |
| | | $F_{N_r,2}^M$ | 0.3 | 22 | 72 | 11 | 9 | 8 | 8 | 0.97 |
| | | $F_N^M$ | 0.3 | 28 | 86 | 13 | 11 | 15 | 8 | 0.97 |
| 50% MVC | **4** | $F_{N_r,1}^M$ | 2.9 | 164 | 320 | 27 | 18 | 33 | 28 | 0.92 |
| | | $F_{N_r,2}^M$ | 1.3 | 96 | 200 | 14 | 4 | 21 | 9 | 0.96 |
| | | $F_N^M$ | 1.4 | 100 | 214 | 15 | 6 | 21 | 8 | 0.95 |
| | **8** | $F_{N_r,1}^M$ | 5.4 | 268 | 415 | 32 | 23 | 35 | 35 | 0.86 |
| | | $F_{N_r,2}^M$ | 5.3 | 262 | 258 | 20 | 16 | 23 | 20 | 0.92 |
| | | $F_N^M$ | 5.4 | 268 | 255 | 16 | 17 | 15 | 17 | 0.90 |

For all three modelling approaches at both contraction levels, $F_{TA}$ was more accurately predicted according to all the validation metrics when the electrode density increased, especially during the ramps of force. For example, at 30% MVC, it was obtained $\Delta_1 = 0.1$ $s$, $\Delta_1^F$ in $11 - 18$ $N$, nRMSE in 6–8%, and $r^2$ in 0.98–0.99 from the 81 discharging MUs identified with 4 mm interelectrode distance, while the 14 MUs identified with 16 mm interelectrode distance returned $\Delta_1 = 0.3$ $s$, $\Delta_1^F$ in $22 - 28$ $N$, nRMSE in 11–18%, and $r^2$ in 0.89–0.98 (Table 4).

As expected from the neural drive estimation in Fig 6, for all the modelling approaches and EMG grid configurations, $F_{TA}$ was more accurately predicted at 30% MVC than at 50% MVC (upper row versus lower row in Fig 6), with two to three times lower nRMSE values, and strongly lower $\Delta_1$ and $\Delta_1^F$ and higher $r^2$ values, respectively (Table 4).

## Discussion

### Summary of the work

This study reports a novel approach, summarized in Fig 1, to develop a MN-driven neuromuscular model, that is controlled at the level of the individual MU with experimental motoneuronal data. This model advances the recently reviewed [17] state-of-the-art of neuromuscular modelling on multiple aspects, and finds later-discussed applications in the complementary fields of motor control and MSK modelling. The population of Hill-type MU actuators (Fig 1F) transforms, with advanced models of the MU's excitation, activation, and contraction dynamics (Figs 3 and 4), a vector of MN spike trains into a vector of transient MU forces that sum across the modelled MU population to yield the whole muscle force (Fig 8). The neuromuscular model and the continuous distribution of the MU's recruitment and force-generating properties across the MU pool were respectively made subject-specific using a subject-specific MSK model derived from medical images (Fig 1H and 1I), and muscle-specific, using results from the literature (Fig 2). The neural control to the MN-driven model is also subject-

specific as the vector of input MN spike trains is derived from HDEMG signals recorded during the participant's voluntary task (Fig 1A and 1B). The experimental vector of $N_r$ identified spike trains resulting from HDEMG signal processing was either directly input to the neuromuscular model of $N_r$ MUs (Fig 1C) or first extrapolated to an estimation of the discharge activity of the complete population of $N$ MNs [41], which was then input to the neuromuscular model of $N$ MUs (Fig 1D). The accuracy in estimating the neural drive to muscle of this novel MN-driven approach was assessed using experimental datasets of varying quality (Fig 6), with the control vectors of $N_r$ and $N$ MN spike trains (Fig 1E). Finally, the whole muscle force predicted by the MN-driven neuromuscular model was validated against an experimental muscle force (Figs 1G and 8), that was estimated from measured joint torque and bEMG signals recorded from agonist and antagonist muscles (Figs 1J and 5).

## Advancing the state-of-the-art in neuromuscular modelling

The neuromuscular model developed in this study was built as a collection of in-parallel MU actuators, which proved beneficial for advancing the recently reviewed [17] state-of-the-art of neuromuscular modelling on several aspects.

For the first time in Hill-type modelling, we controlled the individual MUs of a modelled MU population with a vector of dedicated experimental motoneuronal controls. The vector of MN spike trains obtained from HDEMG signals provided a comprehensive description of both the dynamics of MU recruitment and MU rate coding for the modelled population of MUs. Conversely in EMG-driven models of whole muscle actuators, the recruitment and discharge dynamics of the MU pool are lumped into a single phenomenological macroscopic neural control where they become indistinguishable, that is either obtained from bEMG envelopes [19,20,90] or from the temporal summation of filtered cumulative spike trains and signal residuals from HDEMG signals [22,23,91]. Besides, in multiscale models of single representative MUs, the dynamics of MU recruitment were either overlooked or accounted for with phenomenological models [28–32,36]. A few studies also assigned motoneuronal controls to modelled populations of MUs [36–38,92–94] but used synthetic data and/or phenomenological models, such as Fuglevand's formalism [95], to describe the discharge and recruitment dynamics of the MU pool. Consequently, these models of MU populations were not used in conditions of voluntary muscle contraction and the force they predicted was indirectly validated against results from other models and not against synchronously recorded experimental data like performed in this study. A recent study [96] implemented a neuromechanical model of a population of twitch-type models of MUs controlled by experimental vectors of MU spike trains. This model was tested with limited samples of few experimental MN spike trains (i.e., 2–22 identified MUs per contraction per muscle); such samples are typically not representative of the real neural drive to muscle, which is a key challenge in MN-driven neuromuscular modelling, as discussed in the present study (Fig 6). In this perspective, it is worth highlighting that our study did not perform a calibration step of the muscle model parameters, contrary to [96], to avoid any kind of error cancellation that would non-physiologically correct for this experimental limitation when validating the model. Then, [96] proposes a linear summation of the fusing twitches in a MU, that contradicts some experimental evidence discussed in the present study. The physiological nonlinear summation of the MU twitches during muscle contraction, classically modelled in twitch-type models with a nonlinear gain that is MU-specific and depends on the instantaneous firing rate [95], is in our study predicted by the detailed models of the MU activation dynamics described in Eqs 14–18 (Fig 4G and 4H). Finally, the inter-relation in a MU pool proposed in [96] between MU firing properties, MU twitch contraction time, and MU type, as well as their bimodal distribution in the MU pool, remain

unproven in human muscles with contradictory findings [5,55,84,97], as reviewed [98] and discussed [4] previously. For these reasons, the MN-driven approach developed in the present study is the first that can control with experimental vectors of inputs, that are representative of the real neural drive to muscle, a population of physiologically accurate models of the individual MUs' force-generating dynamics in the forward prediction of human voluntary muscle contraction. Moreover, the proposed MN-driven model while also relies on a detailed description of the excitation-contraction dynamics of the active muscle tissue and on a set of experimental spike trains that is more easily interpretable than the phenomenological neural controls used in single-input multiscale models of whole muscle actuators in the literature.

Discretely sampling the active muscle tissue into individual MUs provided a convenient framework for the precise distribution of the muscle's properties across the MU pool, including the novel experimental distributions in Fig 2 of MU recruitment threshold, maximum isometric forces, and type, some of which were never included in neuromuscular models. This description of the muscle as a cohort of MUs also provided a convenient structure for deriving physiological assumptions and simplifications to the rheological structure of the model (see S1 Text (Section 2) for details).

This MU-scale approach was ideal for developing advanced models of the MU's excitation, activation, and contraction dynamics (Fig 4). Experimental data from the literature was used in this study to derive adequate mathematical equations (Eq 11 to Eq 18), identify fitting physiological parameters (Table 1), and validate the modelled dynamics (Fig 4). Importantly, the identification of the parameter values in Eq 11 to Eq 18 and the validation of those mathematical descriptions were performed using different sets of experimental data from the literature that were measured at different discharge rates (see Appendices A.1 and A.1 for details). Furthermore, we limited the amount of multiscale and inter-species scaling inaccuracy, which is an important limitation of single-actuator Hill-type models [17], by considering source experimental data, in turn and decreasing order of preference, from studies on individual human MUs or bundles of fibres, human fibres and sarcomeres, cat fibres, rodent fibres, rodent muscles, and amphibian fibres or muscles. In doing so, the simulated MN APs, fibre APs, and free calcium twitches were more physiological and consistent with mammalian dynamics than previous approaches based on experimental amphibian data at low temperature (Fig 4B, 4E and 4F). For the first time in Hill-type-like modelling, the dynamics of calcium-troponin concentration in the sarcoplasm were described according to experimental data (Eq 17) (see S1 Text (Section 4) for details) and the free calcium transients were made nonlinearly dependent to both MU length and MU type based on experimental measurements (Eq 14 to Eq 16). Hence, this model further reconciled the phenomenological Hill-type modelling approach with the real physiological mechanisms responsible for muscle force generation it describes.

While this study presents novel techniques to develop state-of-the-art neuromuscular models as MN-driven pools of MU Hill-type actuators, it remains to assess the performance of these more physiological and complex models compared to the more common phenomenological single-input single-actuator approaches, like the standard bEMG-driven Hill-type models [21]. Almost no studies that proposed models of MU pools performed this comparative assessment, which was therefore never reviewed [99]. Only three studies compared, for limited contraction tasks, the force outputs of a model of MU pool and of a single muscle-scale model, using Hill-type [100] and twitch-type [91,96] approaches. [91,100] reached the same conclusions. The models of MU pool and the single-actuator muscle-scale models can yield equally accurate predictions of muscle force in the time domain, although the MU pool model always performs better in the frequency domain. Contrary to single-actuator models, models of MU pool can predict the higher frequency force fluctuations, crucial in the analysis of steadiness for example. This is explained by the superiority as control signals of vectors of MU spike

trains, immune to waveform cancellation, over rectified and smoothed bEMG signals. It is worth noting that the single-actuator Hill-type model in [100] is significantly more complex and elaborate in its phenomenological description of the MU dynamics than traditional EMG-driven Hill-type models, as reviewed [17], while the single-actuator twitch-type model in [91] was fully calibrated to match the force traces. It is therefore possible that MN-driven models of MU pool behave more accurately than more standard single-actuator bEMG-driven approaches, especially between contractions where the neural strategy changes, e.g., between different contraction intensities or when fatigue occurs, considering that calibrated bEMG-driven models are known to behave poorly at other activation levels [101]. In this respect, cumulative spike train driven single-actuator models [22], that describes the MU pool recruitment and firing strategies, were shown to more accurately predict muscle forces than classic bEMG-driven Hill-type approaches.

Consequently, the field requires a systematic assessment of the comparative performance of standard single-actuator neuromuscular models and models of MU pools in a variety of contraction tasks and in generic, subject-specific, and calibrated approaches. Finally, beyond the prediction accuracy, MN-driven models of MU pools provide detailed insights into the muscle's internal dynamics and finds practical applications that standard single-actuator bEMG-driven models cannot provide, as later discussed.

## Modelling limitations

Although the approach presented in Fig 1 provides a state-of-the-art approach for investigating and modelling the dynamics of the MU pool in forward simulations of human voluntary muscle contraction, it suffers from the following modelling limitations. First, the muscle model was simplified to account for a rigid tendon. This approach was acceptable for the TA and decreased the computational load. Yet, the force equilibrium between the elastic tendon and the active muscle tissue should be considered with Eq S9 in S1 Text (Section 2) for muscles with more compliant tendons or higher $\frac{l_s^T}{l_0^M}$ ratios, in which case $\bar{l}$ would vary and the PEE would not be neglected anymore. Second, physiologically describing the TA muscle with a MU resolution required assigning to the MU pool physiological distributions of neuromechanical parameters (Fig 2) obtained from TA-specific experimental measurements from the literature. Although multiple studies also investigated these parameters in the human thenar, ankle extensor, and masseter muscles, those properties remain mostly unknown for the other human muscles, as recently reviewed [57]. Third, the pipeline in Fig 1 was only applied to one subject in this study. Yet, the approach is general and can be reapplied to other subjects following the same method. Fourth, we had to correct a limitation of our reconstruction method [41] for the simulated MUs recruited during the plateau of constant recorded torque. When the discharge frequency of such simulated MU was below 7 Hz in average over the plateau, the predicted MU force was set to zero to avoid adding random noise to the predicted whole muscle force at the highest developed force levels.

## Experimental neural control and prediction accuracy

The accuracy of the whole muscle force predictions in Fig 8 was mainly related to the accuracy of the experimental neural control in approximating the neural drive to muscle (Fig 6). It is currently impossible to identify the discharge activity of the complete MU pool of a muscle with surface EMG because of the filtering effect of the volume conductor. It is however possible to increase both the number of identified MUs and the ratio of identified low-threshold MUs by increasing the size and the electrode density of HDEMG surface grids to obtain

experimental samples of identified MUs that are representative of the discharge activity of the complete MU pool [40].

The experimental sample of $N_r$ = 81 MUs, that was obtained at 30% MVC with a dense and large grid of 256 electrodes with 4 mm interelectrode distance, was representative of the discharge activity of the complete MU pool because the identified MUs both spanned across the entire recruitment range, with the lowest-threshold identified MU recruited at 0.6% MVC, and were homogeneously distributed across the recruited MU pool, with 12% to 20% of the identified MUs in each of the 5% ranges sampling the recruitment range, as displayed with the histogram in Fig 6C. Consequently, the neural drive to muscle was accurately estimated with the discharge activity of the $N_r$ = 81 MUs (Fig 6C and Table 3), which produced the most accurate force estimations in this study when input to the neuromuscular model (Fig 8C and Table 4). Consistent with previous conclusions [40], the experimental samples obtained at 30% MVC with lower electrode density (8 and 12 mm interelectrode distance) provided a less accurate description of the discharge activity of the complete MU pool. At this force level, increasing the electrode density better samples the action potential profiles of the low-threshold MUs across multiple electrodes, enabling their identification. In these datasets, although the identified MUs also spanned across the entire recruitment range, fewer MUs were identified and their distribution across the MU pool shifted towards relatively larger sub-population of high-threshold MUs (histograms in Fig 6A and 6B). Consequently, the neural drive to muscle (Fig 6A and 6B and Table 3) and the predicted muscle force (Fig 8A and 8B, higher $nRMSE_{r1}$ and $nRMSE_{r2}$ values in Table 4) were underestimated in the regions of low-force generation when low-threshold MUs are recruited. This experimental under-representativity of the low-threshold MUs was corrected using a computational method [41] by deriving the continuous distribution of the MN's electrophysiological properties across the entire MN pool and reconstructing the discharge activity of the MUs that were not identified experimentally (Fig 1D). The reconstruction method populated the two experimental samples with simulated populations of low-threshold MUs, in which case the accuracy in predicting both the neural drive (blue dotted versus solid red traces in Fig 6A and 6B) and the muscle force (Fig 8A and 8B) systematically increased (Tables 3 and 4). As expected, this reconstruction step did not improve the accuracy of the predictions for the experimental sample of $N_r$ = 81 MUs, which was already representative of the discharge activity of the MU pool.

The discharge activity of the low-threshold MUs was not identified during high-force contractions up to 50% MVC, even with high electrode density. Classically, the decomposition algorithms converge towards the large MUs that contribute most to the EMG signals, while action potentials of the small MUs of lower energy are masked by the potential of larger units, more of which are recruited at 50% MVC than at 30% MVC [40]. Beyond shifting the MU identification towards the highest-threshold MUs and yielding imbalanced MU distributions (histograms in Fig 6D and 6E), the two experimental samples of identified MUs at 50% MVC did not span across the entire recruitment range, the identified MUs being first recruited above 5% and 20% MVC. Consequently, those experimental samples inaccurately predicted the onsets of the neural drive (Fig 6D and 6E) and of the whole muscle (Fig 8D and 8E) with 1.2 to 5.2 seconds delay and initial underestimations of the muscle force up to 268 N ($\Delta_1$ and $\Delta_1^F$ in Table 4). Because the reconstruction method relies on the neural drive estimated from the experimental sample (Fig 6D and 6E), it could not correct for this source of inaccuracy (solid red traces in Figs 6 and 8), although it improved the predictions by better distributing the MU's discharge activity across the identified MU pool, as discussed. It is worth noting that low-threshold MUs are usually not identified with grids of low electrode density at 30% MVC either, as hinted with the histograms in Fig 6A and 6B, and the resulting experimental samples

of MUs hence identified usually do not span across the entire recruitment range, yielding similar prediction inaccuracies. To address this limitation when controlling single whole muscle actuators with HDEMG signals, previous studies [22,23,91] summed the experimental cumulative spike train with the EMG signal residual that was not explained by the identified MU spike trains to approximate the neural control to muscle. Consequently, the accuracy of the predictions made by the neuromuscular model developed in this study are sensitive to the number, the span, and the distribution across the MU pool of the experimental sample of identified MUs, which can, to some extent, be extrapolated to the discharge activity of the complete MU pool using our published reconstruction method [41].

As displayed in Fig 8A–8E, the accuracy of the predicted forces $F^M(t)$ also depended on the assignment of the maximum MU isometric forces $f_{0,k}^{MU}$ to the modelled MU pool. The $f_{0,k}^{MU}$ values were obtained from the muscle-specific $f_0^{MU}(k)$ distribution in Eq 8, which was directly mapped to the $N$ modelled MUs when the complete MU pool was reconstructed. This physiological approach implemented the continuous distribution of the MU forces across the MU pool (Fig 7D) and almost systematically returned the most accurate $F^M(t)$ predictions (Fig 8A–8E and Table 4). When the muscle was described as populations of $N_r$ MUs controlled by the experimental vector of MN spike trains, representative $f_{0,k}^{MU}$ values were derived from the $f_0^{MU}(k)$ distribution and assigned to the experimental MUs to account for the force-generating properties of the MUs not identified experimentally. In such case, the distribution of simulated MU forces (Fig 7B and 7C) was not physiological and interpretable. We showed that locating the $N_r$ MUs into the real $T^{th}$-ranked MU pool with Eq 6 before assigning representative $f_{0,k}^{MU}$ values (Fig 7C) corrected for the non-homogeneous distribution of the experimental samples across the MU pool, which was previously discussed, and allowed predictions accuracies of $F^M(t)$ close to those obtained with the complete population of $N$ MUs (green dashed versus solid red traces in Fig 8 and Table 4). For example, the few low-threshold MUs identified with grids of low electrode density were assigned high representative $f_{0,k}^{MU}$ values to represent the force-generating properties of the large population of small MUs that were not identified experimentally. When the representative $f_{0,k}^{MU}$ values were assigned under the assumption of a homogeneous distribution of the $N_r$ MUs across the MU pool (Fig 7B), the predictions of the whole muscle force were systematically less accurate, again under-evaluating the force-generating activity of the low-threshold MU population.

## Interfacing the fields of subject-specific motor control, neuromuscular modelling, and MSK modelling

The neuromuscular model developed in this study brings together state-of-the-art experimental and modelling techniques from the three complementary fields of motor control, neuromuscular modelling, and MSK modelling.

The subject-specific neural control was obtained from recently developed experimental and processing techniques (Fig 1A–1C) that yielded the identification of much larger samples of discharging MUs from recorded HDEMG signals than commonly obtained in the literature [11], i.e., up to 81 MUs in the TA muscle, and high ratios of low-threshold MUs [40].

The neuromuscular model, the modelling novelty of which was discussed previously, was scaled with the muscle-specific distribution of MU maximum isometric forces $f_0^{MU}(k)$ derived from the literature (Fig 2) and from the muscle architectural parameters (Table 2), that were obtained from a subject-specific MSK model (Fig 1H and 1I) built from the segmentation of MRI scans using state-of-the-art automated tools [50,51]. MSK model predictions can be sensitive to the uncertainties in parameter identification and MSK architecture [102,103]. The

subject-specific properties of the TA muscle used in this study were compared to those from a generic published model [54], and the same model scaled to the anthropometry of the subject with the Opensim built-in tools [48,49]. The values of the optimal length $l_0^M$ and tendon slack length $l_s^T$ did not vary between the three models (<3% variation). The subject-specific muscle-tendon length $l^{MT}$ at 30˚ plantarflexion was 3% shorter and 12% longer than in the scaled and generic models, respectively. Considering the simplification of rigid tendon, these differences in length linearly propagate to the estimation of the normalized MU length $\bar{l}$ and nonlinearly to the FL and length-dependent activation dynamics of the MUs (Fig 4). The subject-specific MRI-based maximum isometric force $F_0^M$ was 15% lower than proposed in the generic model. This difference linearly propagates to the linear scaling of the distribution of MU isometric forces $f_0^{MU}$ in Eq 8 (Fig 7B–7D) and to the amplitude in N of the predicted whole muscle force (Fig 8). The highest difference was obtained for the muscle moment arm with the ankle joint in dorsiflexion, where the subject-specific quantity was 33% and 40% lower than the scaled and generic ones. This difference linearly propagates to the estimation of experimental muscle force from experimental joint torque (Fig 5) that was used for model validation (Fig 8).

The model proposed in this study and the predictions displayed in Fig 8 did not rely on any parameter calibration that would minimize external cost functions, like commonly proposed in EMG-driven approaches where some subject-muscle-specific properties are, for example, calibrated by minimizing the difference between predicted and experimental joint torques [20,21].

Beyond the MSK parameters directly measured with the subject-specific MSK model, it is worth noting that the remaining parameters in the equations that define the model's force-generating properties (Eq 8 and Eq 11 to Eq 18) are fitting experimental data from the literature, that are independent from the EMG and torque signals that were used in this study to control and validate the model.

With this subject-specific and MN-driven approach, the accurate prediction of the whole muscle force amplitude at 30% MVC in Fig 8E–8G were consequently obtained without parameter calibration owing to a physiological and comprehensive description of the neural drive (Fig 6), which the bEMG envelopes normalized by MVC signals cannot achieve, an adequate distribution of the MU's maximum isometric forces $f_{0,k}^{MU}$ across the modelled MU population (Fig 7), and an estimation of muscle co-contraction during ankle dorsiflexion (Fig 5), as discussed.

In cases for which subject-specific measurements are not possible or muscle-specific data are not as available in the literature as for the TA, which is the case for most human muscles [57], missing muscle architectural parameters or the coefficients in Eq 8 could be included in an optimization routine aiming to minimize the difference between experimental and predicted joint torques, for example.

The only calibrated parameter in the pipeline in Fig 1 was the size of the experimentally identified MNs, that was required for the specific test case of complete reconstruction of the MN pool (Figs 6 and 8, red traces) to derive the continuous distribution of electrophysiological properties across the population of MN LIF models. Again, this parameter identification was independent from the experimental measurements of joint torque used in this study to validate the model. The two other approaches taken to predict the muscle force solely relied on the experimental samples of identified MNs (Fig 8, blue and traces) and did not require the calibration of the MN size parameter.

## Applications and future perspectives

The MN-driven multi-MU model proposed in this study comprehensively describes the recruitment and firing strategy of the MU pool during human voluntary muscle contractions.

For these reasons, this detailed model finds applications in answering scientific questions where the lump dynamics of the classic single-actuator bEMG-driven Hill-type models provide limited help.

For example, current research [42,43] aims for a new generation of high-performance multi-articulating prosthetic limbs, which, besides relying on direct skeletal attachment via osseointegration, muscle reinnervation, and implanted sensors, also require advanced MU-based algorithms, like the one proposed in this study, that implements the mechanistic relationship between the individual motoneuronal activity and motor function.

In the field of biomechanics, the presented MU-based model can actuate the recently developed volumetric representations of muscles in MSK models [51] by mapping the modelled population of MUs to the volumetric population of lines of action, that are consistent with the segmented muscle geometry. Volumetric representations of Hill-type-actuated muscle models have only rarely been proposed in the literature [36,104]. Beside addressing the current limitations of modelling muscles as single rectilinear segments, this volumetric mapping would also provide a solution to the indeterminacy in in S1 Text (Section 2) by assigning MU-specific lengths $\overline{l_k}$ to the modelled MU actuators and would shift the current approach towards a more physiological nonlinear summation of the MU forces based on muscle architecture. Modelling the interaction between individual MU lines of action with the tendon and skeletal structures would further bridge the gap in modelling the interplay between motor control and resulting human motion.

In the study of the human muscle architecture, by combining the aforementioned mapping of the MU pool in volumetric muscle representations with recently developed ultrasound measurement techniques for tracking MU twitches and mechanical properties [34,105], one could gain insights into the distribution of MU territories within human muscles, which remains an open question. The volumetric mapping of MUs also provides a convenient framework for modelling the transversal mechanical interactions between MUs [106], and the resulting force-varying load path within the muscle tissue, that results in a nonlinear summation of the individual MU forces.

In the field of neurophysiology, the MN-driven model of MU population proposed in this study is suited for integrating MN synergies in simulations. Contrary to single-input Hill-type models that lump the dynamics of synaptic input, MN recruitment, and MN discharge into a single phenomenological signal, the MU pool modelled in Fig 1 can be divided into functional clusters and receive different common inputs to reduce the dimensionality of the control [35]. For example, in multi-muscle MSK models with MU-actuated volumetric representations of muscles, the model proposed in Fig 1 is also suited for the investigation of neural synergies between muscles, where MN clusters span across muscles [35], with strategies specific to the muscle groups [107]. In the field of neurophysiology, the model of MU population proposed in this study is also applicable for the computational generation of surface EMG signals during voluntary movement [108].

In the study of human muscle contraction dynamics, the model developed in this study provides a credible window onto the distribution of the excitation-contraction dynamics of individual MUs across the MU pool (Fig 4) that cannot be measured in human *in vivo*, and advances our understanding of the muscle-specific neuromechanical strategies for muscle force generation. For example, Fig 4 suggests that, at 30% MVC, the highest-threshold recruited MUs with the largest force-generating capacities are not those producing the highest forces within the recruited pool due to low activation states explained by the onion skin theory (Fig 8A and 8D). As the model integrates realistic descriptions of the MU-specific firing, recruitment, excitation, and activation dynamics responsible for muscle force generation, it also provides a convenient

framework for integrating and investigating MU-related mechanisms, such as the effect of fatigue on the MU firing and recruitment strategies [109] or other fatigue-induced changes in MU activation dynamics as already proposed in single-MU models [110,111].

However, some currently investigated limitations can prevent the proposed MN-driven MU-based model from being readily applicable to the aforementioned scientific questions. First, although the pipeline in Fig 1 is fully suited for dynamic contractions, the study proposed here was constrained to isometric tasks due to current experimental limitations in the decomposition of HDEMG signals into MU spike trains during motion. New computational approaches are emerging to identify individual spike trains in quasi-static [112] and dynamic [113] tasks, where 7–20 MUs can be currently identified. Second, data acquisition remains challenging and time-consuming. As opposed to straightforward bEMG recording and filtering, the pipeline in Fig 1 requires decomposing HDEMG signals and manually editing the identified spike trains editing, while taking precautions to identify the full spectrum of discharging MUs for accurate predictions (Fig 6), as discussed. These challenges should be addressed with the rapid emergence of guidelines for manufacturers on HDEMG grid design [40], open-source tools for automatic HDEMG decomposition and spike trains edition [114], automated spike train identification approaches based on machine learning [115] and blind-source separation methods [116], and MU pool reconstruction methods [41].

Finally, with four ODEs to solve at each time step for each of the modelled MUs, the model proposed in this study is currently computationally too expensive for real-time applications, as opposed to the classic single-input Hill-type actuators, which usually include zero to two ODEs. For the simulation of a 40s-long muscle contraction using a standard laptop (RAM: 12 GB, CPU: one Intel Core i7-1165G7 2.80 GHz), the pipeline in Fig 1 implemented in Python ran in 18 minutes for a MU pool of $N_r$ = 16 MUs (Fig 8A, dotted trace), in 63 minutes for $N_r$ = 81 MUs (Fig 8C, dotted trace), and in 240 to 300 minutes for the complete pool of $N$ = 400 MUs (Fig 8, red traces), that additionally required preliminary MU pool reconstruction. For comparison, a single-actuator rigid-tendon Hill-type model [60] ran in less than a minute on the same machine for the same 40s-long trapezoidal isometric contraction where the HDEMG signals were averaged and normalized to MVC signals. Besides implementations of the model in a faster compiled programming language (for example C++) and using better performing machines, the computational speed could be drastically increased by using parallel computing to solve the independent MU dynamics, by considering smaller reconstructed populations of 50 MUs without loss of accuracy in the estimated neural drive to muscle, as previously discussed [41], or by, for example, simplifying the current model of MU excitation and activation dynamics with analytical descriptions of the fibre APs and calcium transients instead of the two 2$^{nd}$-order ODEs in Eq 13 and Eq 14.

## Conclusion

We developed the first MN-driven neuromuscular model of a population of individual Hill-type MUs controlled by a vector of dedicated experimental motoneuronal controls (Fig 1). The model distinguishes the dynamics of MU recruitment from rate coding and produces the whole muscle force as the summation of the forces generated by the individual modelled MUs. The model is subject-specific (Table 2), muscle-specific (Fig 2), and includes an advanced and physiological model of the MUs' activation dynamics (Figs 3 and 4 and Table 1). The motoneuronal controls, derived from HDEMG signals, are experimental and decode the subject's intention, which makes the neuromuscular model applicable to the simulation and investigation of human voluntary muscle contraction. The model's predictions of the whole muscle force are sensitive to the quality of the experimental neural control. Accurate force predictions

were obtained when the effective neural drive to muscle was accurately estimated from the decoded MN spike trains (Figs 6C and 8G), i.e., when the experimental samples of identified MUs were representative of the discharge activity of the complete MU pool. This was obtained when the muscle's myoelectric activity was recorded with large and dense grids of EMG electrodes during medium-force contractions, in which case the identified MUs span across the complete range of recruitment and are homogeneously distributed across the MU pool (Figs 6 and 8). Otherwise, the discharge activity of the low-threshold MUs is typically not identified, especially during high-force contractions, and the force predictions are inaccurate in the regions of low-force generation (Table 4). Inferring with a computational method the discharge activity of those MUs that were not identified experimentally improves the results to some extent and provides a window onto the continuous distribution of the MUs' force-generating dynamics across the MU pool. The accuracy of the force predictions also relies on a physiological assignment of the MU-specific force-generating properties to the modelled population of Hill-type MUs (Fig 8B–8D). This MN-driven model advances the state-of-the-art of neuromuscular modelling, brings together the interfacing fields of motor control and MSK modelling, and finds applications in numerous fields, including the investigation of the human neuromuscular dynamics during voluntary contractions, neural synergies, and human-machine interfacing. The implementation of the method is publicly available at https://github.com/ArnaultCAILLET/MN-driven-Neuromuscular-Model-with-motor-unit-resolution. The segmented medical images and the subject-specific MSK model are publicly available at https://zenodo.org/records/10069266.

## Supporting information

**S1 Text. Supporting methods and information for the derivation and the validation of the neuromuscular model.**
(PDF)

## Author Contributions

**Conceptualization:** Arnault H. Caillet, Andrew T. M. Phillips, Dario Farina, Luca Modenese.

**Data curation:** Arnault H. Caillet, Luca Modenese.

**Formal analysis:** Arnault H. Caillet.

**Funding acquisition:** Dario Farina.

**Investigation:** Arnault H. Caillet, Luca Modenese.

**Methodology:** Arnault H. Caillet, Andrew T. M. Phillips, Dario Farina, Luca Modenese.

**Resources:** Dario Farina.

**Software:** Arnault H. Caillet, Luca Modenese.

**Supervision:** Andrew T. M. Phillips, Dario Farina, Luca Modenese.

**Validation:** Arnault H. Caillet.

**Visualization:** Arnault H. Caillet.

**Writing – original draft:** Arnault H. Caillet.

**Writing – review & editing:** Arnault H. Caillet, Andrew T. M. Phillips, Dario Farina, Luca Modenese.

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
