## [Decision Letter · Decision Letter 0]

31 Aug 2023

Dear Dr. Modenese,

Thank you very much for submitting your manuscript "Motoneuron-driven computational muscle modelling with motor unit resolution and subject-specific human musculoskeletal anatomy" for consideration at PLOS Computational Biology. As with all papers reviewed by the journal, your manuscript was reviewed by members of the editorial board and by several independent reviewers. The reviewers appreciated the attention to an important topic. Based on the reviews, we are likely to accept this manuscript for publication, providing that you modify the manuscript according to the review recommendations.

Please note in particular that, although both reviewers appreciated the thorough approach in the presented muscle model, they felt that there should be more discussion of how it fits into the context of existing models. Under what circumstances would this model (rather than a simpler one) be most applicable? How do the results compare to the classic single-input Hill-type model and what is the relative computational cost?

Sincerely,

Barbara Webb

Academic Editor

PLOS Computational Biology

Marieke van Vugt

Section Editor

PLOS Computational Biology

Reviewer's Responses to Questions

**Comments to the Authors:**

Reviewer #1: The manuscript presents the detailed development and validation of a motoneuron driven, physiologically accurate, muscle model. Subject and muscle specific parameters were derived from medical imaging and literature data to generate an accurate representation of the individual neuromuscular anatomy and physiology. Motoneuron spike trains were experimentally derived from measurement of the tibialis anterior muscles during submaximal voluntary isometric contraction, and used to drive and validate the proposed model. Three different grid configurations, with varying density of HDEMG, were evaluated to assess the effect of experimental recording quality on the TA force prediction. The verification of the modelling pipeline was detailed and provided convincing results that supported the conclusions of the manuscript.

Overall, this paper is carefully written and a wonderful example of how a computational model should be developed and validated. Every modelling choice was planned, justified, and supported by literature. I could not fault any part of the modelling that was not already detailed in the limitations. The well organised equations also provide an excellent summary of much of the cited literature, which was needed. Finally, I truly appreciated that the authors provided the modelling code, which will surely boost future impact.

Generic comments

1. While the paper is technically excellent, I believe the parts of the discussions related to future applications were a bit vague and generic. The number of required parameters, the challenges of data acquisition, and the inability to function in dynamic conditions might limit the future use of the proposed model for most biomechanics applications. Overall, I would appreciate to see the discussion extended to highlight and detail how future applications or scientific investigations could truly leverage this model.

2. As the focus of the paper was on muscle force, one is left wondering how this modelling approach would compare to a classic single-fibre Hill-type bEMG-driven muscle model. An additional analysis of this type would be very useful.

Specific comments

P10 line 186. It is unclear whether duplicates of motor units were automatically or manually removed.

P13 line 260. Mus should be Mus

P15 line 323-324. The state variables have not been previously stated.

P38 line 837. “true neural to muscle” missing a word?

Reviewer #2: The authors have developed a complex computational model, supported by experimental work, that is controlled at the level of individual motor units (MUs) and accounts for the dynamics of MU recruitment.

The authors include some subject-specific metrics (muscle volumes, centroid lines of action) extracted from MRI. How big an effect do these subject-specific parameters have on the predicted results, as compared to generic values (i.e., how much value does this additional customization add to the model performance)?

Parameters of the model are derived from a combination of calibration of some parameters, and direct measurement and application of others. As such, a true model validation has not been performed. Please include discussion of this limitation in the manuscript.

Lines 621-622: What was the reason for these inaccurate description of discharge activity?

Why were experimental samples better approximated at lower MVCs (30% versus 50%)?

It would be informative if the authors discussed applications where they expect their model to perform better than current models, and why (e.g., phenomenological models).

The authors do not provide any benchmarking against more simplistic models, therefore it is difficult to assess if the additional complexity is warranted, or if the additional effort and complexity required has a negligible outcome on model performance.

Please include some numerical values to describe the computational costs associated with the model (i.e., simulation time for a specific set of hardware) as compared to the classic single-input Hill-type actuators.

**Have the authors made all data and (if applicable) computational code underlying the findings in their manuscript fully available?**

Reviewer #1: Yes

Reviewer #2: Yes

PLOS authors have the option to publish the peer review history of their article (what does this mean?). If published, this will include your full peer review and any attached files.

Reviewer #1: No

Reviewer #2: No

Figure Files:

Data Requirements:

Reproducibility:

References:

---

## [Decision Letter · Decision Letter 1]

16 Oct 2023

Dear Dr. Modenese,

We are pleased to inform you that your manuscript 'Motoneuron-driven computational muscle modelling with motor unit resolution and subject-specific musculoskeletal anatomy' has been provisionally accepted for publication in PLOS Computational Biology.

Best regards,

Barbara Webb

Academic Editor

PLOS Computational Biology

Marieke van Vugt

Section Editor

PLOS Computational Biology

Reviewer's Responses to Questions

**Comments to the Authors:**

Reviewer #1: I would like to thank the authors for their extensive replies to my comments.

Reviewer #2: The authors have provided a detailed and thorough response to the comments previously raised. I have no further comments.

**Have the authors made all data and (if applicable) computational code underlying the findings in their manuscript fully available?**

Reviewer #1: Yes

Reviewer #2: Yes

PLOS authors have the option to publish the peer review history of their article (what does this mean?). If published, this will include your full peer review and any attached files.

Reviewer #1: No

Reviewer #2: No

---

## [Editor Report · Acceptance letter]

13 Nov 2023

PCOMPBIOL-D-23-01007R1 

Motoneuron-driven computational muscle modelling with motor unit resolution and subject-specific musculoskeletal anatomy

Dear Dr Modenese,

I am pleased to inform you that your manuscript has been formally accepted for publication in PLOS Computational Biology. Your manuscript is now with our production department and you will be notified of the publication date in due course.

With kind regards,

Anita Estes
